



# Estimation of the error covariance matrix for IASI radiances and its impact on ozone analyses

Mohammad El Aabaribaoune[1], Emanuele Emili[1], and Vincent Guidard[2]

[1]CECI, Université de Toulouse, Cerfacs, CNRS, Toulouse, France
[2]CNRM, Météo France, Toulouse, France

**Correspondence:** M. El Aabaribaoune (elaabaribaoune@cerfacs.fr)

**Abstract.** In atmospheric chemistry retrievals and data assimilation systems, observation errors associated with satellite radiances are chosen empirically and generally treated as uncorrelated. In this work, we estimate inter-channel error covariances for the Infrared Atmospheric Sounding Interferometer (IASI) and evaluate their impact on ozone assimilation with the chemical transport model MOCAGE (MOdèle de Chime Atmospheric à Grande Echelle). The method used to calculate observation

errors is a diagnostic based on the observation and analysis residual statistics already adopted in numerical weather prediction centers. We used a subset of 280 channels covering the spectral range between 980 and 1100 cm$^{-1}$ to estimate the observation error covariance matrix. We computed hourly 3D-Var analyses and compared the resulting $O_3$ fields against ozonesondes and the measurements provided by the Microwave Limb Sounder (MLS).

The results show significant differences between using the estimated error covariance matrix with respect to the empirical

diagonal matrix employed in previous studies. The validation of the analyses against independent data reports a significant improvement, especially in the tropical stratosphere. The computational cost has also been reduced when the estimated covariance is employed in the assimilation system.

## 1 Introduction

Ozone is an important trace gas that plays a key role in the Earth's radiative budget (Iglesias-Suarez et al., 2018), in the chemi-

cal processes occurring in the atmosphere, and in the climate change (United Nations Environment Programme [UNEP] 2015). Tropospheric ozone also behaves as a pollutant with negative effects on vegetation and human health (UNEP, 2015). The stratospheric ozone is, nevertheless, a vital component of life on the Earth since it protects the biosphere from harmful ultraviolet solar radiation (WMO, 2014). Therefore, monitoring the atmospheric ozone has been a subject of numerous research studies and projects (e.g. Monitoring Atmospheric Composition and Climate (MACC) project (Inness et al., 2013) ). $O_3$ surveillance

is carried out through numerical forecast models and observational systems. The information arising from these two sources is, thereafter, combined with the data assimilation techniques to improve the system state and forecasts.

Remote sounding from satellites is an essential component of the observation's network (Clerbaux et al., 2009). Several remote sensors relying on thermal emission of the Earth and the atmosphere have demonstrated their ability to provide appropriate information for total columns or vertical profiles of atmospheric gases such as water vapor, carbon dioxide, and ozone





(Clarisse et al., 2008). Furthermore, the role of thermal infrared sounders does not typically end at the monitoring atmospheric gases, a large number of applications have taken advantage of these measurements: meteorological parameters (clouds, temperature, and humidity), climate change (e.g., (MacKenzie et al., 2012)). Infrared Atmospheric Sounding Interferometer (IASI) is one of these thermal infrared sounders onboard Metop which provides global scale observations for a series of key atmospheric

species (Clerbaux et al., 2009).

Data assimilation has been introduced relatively recently in atmospheric chemistry, in the stratosphere layer (Fisher and Lary, 1995) and for the troposphere (Elbern, H., Shmidt H. and A., 1997). Chemical fields estimated by chemical transport models (CTM) are combined with observations to construct a realistic picture of the atmospheric composition evolution (Lahoz et al., 2007). Numerous studies have been conducted assimilating satellite retrievals of ozone (Emili et al., 2014; Massart et al., 2009).

However, the quality of analyses might be influenced by the prior information used for the retrievals. Recent studies (Emili et al., 2019) attempted to assimilate satellite radiances directly in a chemical transport model (CTM) to overcome this issue. In chemical assimilation systems that assimilate radiances directly, but also in most of the current satellite retrieval algorithms (Dufour et al., 2012), the observation errors are empirically adapted from the nominal instrumental noise and assumed to be uncorrelated. This assumption is questionable since we use a radiative transfer model that may introduce similar errors among

different spectral channels (Bormann et al., 2009). In other words, an error dependency between channels of the band used is likely to be introduced. Bormann et al. (2009) has listed some other sources of error that could introduce a correlation between the channels or observations. The error can also arise from the instrument calibration and some practices of quality control in the data assimilation system. Liu and Rabier (2003) have shown that the assimilation can lead to sub-optimal analysis errors when observation error correlations are neglected.

The weight given to the observation in the analysis is determined by its error covariance matrix $\mathbf{R}$. Therefore, its estimation plays a crucial role in the assimilation results. While most chemical assimilation systems assume the observation error to be uncorrelated, Numerical Weather Prediction (NWP) systems have estimated non-diagonal observations error covariances for satellite instruments such as Atmospheric Infrared Sounder (Garand et al., 2007; Bormann et al., 2010), IASI (Weston et al., 2013; Bormann et al., 2010; Stewart et al., 2009) and the Spinning Enhanced Visible and Infrared Imager (Waller et al., 2016).

The results found in the literature for the meteorological applications incite us to account for a correlated observation error for the chemical assimilation system as well. Indeed, the studies mentioned above (Weston et al., 2013) show that the inter-channel observation errors are correlated and taking such correlated errors into account in the assimilation leads to improved analysis accuracy. Furthermore, Emili et al. (2019) has highlighted some issues when assimilating radiances in a chemical transport model (increase of the errors compared to the control simulation at some specific altitudes), which might be due to

too simplistic observation errors. The main objective of this study is, thus, to estimate the observation error covariances for IASI ozone-sensitive channels and to evaluate their impact on the analysis accuracy.

The estimation of $\mathbf{R}$ is not straightforward, but a number of statistical methods are already evaluated in the literature. Desroziers et al. (2006) have proposed an estimation based on the observation and analysis residual statistics. By assuming gaussian errors and no correlations between observation and background errors, the error covariance matrix is provided by

the statistical average of observation-minus-background times the observation-minus-analysis residuals. This method has been





used in many studies to estimate the observation errors and inter-channel error correlations (Garand et al., 2007; Weston et al., 2013).

In the present work, we estimate observation errors and their inter-channel correlations for IASI using Desroziers statistics. We evaluate, then, their impact on ozone assimilation in a chemical transport model (MOCAGE). The paper is organized as

follows. The chemical transport model, the radiative transfer model, the assimilation algorithm, the data, and the experimental framework are described in section 2. The estimation of **R** is discussed in section 3. Then, the impact on data assimilation is reported in section 4. Finally, the summary and conclusions are given in the last section.

## 2 Methods and data

### 2.1 Methods

#### 2.1.1 Chemical Transport Model

MOCAGE (MOdèle de Chimie Atmosphérique à Grande Echelle) is the Chemical Transport Model (CTM) used in this study. It is a three-dimensional CTM providing the space and time evolution of the chemical composition of the troposphere and the stratosphere. Developed by Centre National de Recherches Météorologiques (CNRM) at Météo France (Josse et al., 2004), it was used for a large number of applications such as satellite ozone assimilation (Massart et al., 2009; Emili et al., 2014),

climate (Teyssèdre et al., 2007) and air quality (Martet et al., 2009). MOCAGE provides a number of optional configurations with varying domains, geometries and resolutions, as well as multiple chemical and physical parametrization packages.

A global configuration with a horizontal resolution of 2° and 60 hybrid levels from the surface to 0.1 hPa was used. The vertical resolution goes from about 100 m in the boundary layer, to about 500 m in the free troposphere and to almost 2 km in the upper stratosphere. MOCAGE is fed with meteorological fields from numerical weather prediction models such as Météo-

France global model ARPEGE (Action de Recherche Petite Echelle Grande Echelle, (Courtier et al., 1991)), limited area model AROME (Application de la Recherche à l'Opérationnel à Méso-Echelle), and ECMWF NWP model and assimilation system (Integrated Forecast System, IFS) for air quality predictions and ARPEGE-Climat (Déqué et al., 1994) for climate simulations. In our study, the ECMWF IFS meteorological forecasts fields are used. For the chemical scheme, we adopted RACMOBUS which bundle the stratospheric scheme (Lefèvre et al., 1994) and the tropospheric scheme (Stockwell et al., 1997) including

about 100 species and 300 chemical reactions.

#### 2.1.2 Radiative Transfer Model

Remote sensing instruments measure, within a certain wavelength range, the intensity of electromagnetic radiation passing through the atmosphere (radiances). Radiative transfer models are used to simulate the radiation measured by the satellite as a function of atmospheric state, to be able to compare the model state to the observed radiances.

In our study, IASI radiances are simulated using the radiative transfer model RTTOV (Radiative Transfer for TOVS), which was developed initially for the TOVS instrument (R. Saunders and Brunel, 1999). Starting from an atmospheric vertical profile,





RTTOV simulates radiances in the infrared and microwave spectrum. For IASI, it can reproduce radiances with an accuracy of less than 0.1 K (Matricardi, 2009). In this paper, we use the same version used by Emili et al. (2019), i.e. version 11.3 (Saunders et al., 2013). The radiative transfer computations are performed in clear-sky conditions and aerosols were neglected. The surface skin temperature, 2 m temperature, 2 m pressure, and 10 m wind vector are taken from IFS forecasts. The land

surface emissivity is based on the RTTOV monthly TIR emissivity atlas (Borbas and Ruston, 2010) and the Infrared Surface Emissivity Model (ISEM) (Sherlock, 1999) is used over the sea.

### 2.1.3 Assimilation algorithm

The variational data assimilation system of MOCAGE was developed jointly by CERFACS and Météo France in the framework of the European project ASSET (ASSimilation for Envisat data) (Lahoz et al., 2007). It has been used in several studies such

as chemical data assimilation research (Emili et al., 2014; Massart et al., 2009), aerosols data assimilation (Sič et al., 2015) and tropospheric-stratospheric exchange using data assimilation (El Amraoui et al., 2010). The MOCAGE data assimilation system is flexible and allows multiple assimilation options, for example, the choice of the variational method (3D-Var, 4D-Var), the representation of the background error covariance, and the type of observation assimilated. It is also used to produce operational air quality analyses for the European Project CAMS (Marécal et al., 2015).

The background error covariance matrix is divided into two distinct parts, the diagonal matrix of the standard deviations and the correlation matrix. The latter, allowing to spatially smooth the assimilation increments, is modeled through a diffusion operator (Weaver and Courtier, 2001).

In the data assimilation system of MOCAGE, the observation error covariance matrix can be read from the data file previously defined. In the case of diagonal matrix, the variances can be calculated as a percentage of the observation values.

The 3D-Var implementation has been used with hourly assimilation windows. The variational cost function is minimized using the BFGS (Broyden– Fletcher–Goldfarb–Shanno ) algorithm (Liu and Nocedal, 1989). The control vector includes the Skin Surface Temperature (SST) and the ozone.

As we mentioned before, the aim of this work is to evaluate the impact of the estimated observation error on the analysis. Hence, in order to be able to compare our results to those that have been already discussed and validated, we kept exactly

the same configurations as those used in Emili et al. (2019) in terms of model, radiative transfer, and assimilation algorithm parameters. The summary of these configurations is reminded in table 1.





| Parameter | Configuration in the assimilation system |
|---|---|
| Radiative transfer model | RTTOV v11.3 |
| Assimilation algorithm | 3D-Var |
| Spectral window | 980-1100 cm$^{-1}$ |
| Observation error covariance | Both Desroziers statistics and the setup of Emili et al. (2019) |
| Control vector | O$_3$ and SST |
| O$_3$ background error covariance | 3-D-hourly (standard deviation), parameterized (correlations) |
| SST prior information | ECMWF IFS analysis |
| SST background error standard deviation | 4 °C |
| T, H$_2$O fields | ECMWF IFS analysis |
| IR Emissivity | TIR atlas emissivity over land and ISEM model over sea |

Table 1 : Summary of the the configuration of MOCAGE assimilation system.

## 2.2 Data

### 2.2.1 IASI

IASI is one of the instruments operating onboard the polar-orbiting satellite Metop-A launched by the European organization for the Exploitation of Meteorological Satellites (EUMETSAT). It is based on Fourier Transform Spectrometer (FTS) and measures the spectrum emitted by the Earth-atmosphere system in the spectral range between 645 and 2760 cm$^{-1}$ (3.62 and 15.5 μm) with a resolution of 0.5 cm$^{-1}$ after apodization, with a spectral sampling of 0.25 cm$^{-1}$. IASI scans the Earth up to an angle of 48.3° on both sides of the satellite track. The cross-track is observed in 30 successive elementary fields of view,

each composed of 4 instantaneous fields of view corresponding to a 12 km of diameter footprint on the ground (Clerbaux et al., 2009). The swath width on the ground is 2200 km which provides global Earth coverage twice a day. The measurements provide information on atmospheric chemistry compounds such as O3, surface properties (Skin Surface Temperature SST), and meteorological profiles (humidity and temperature).

For this study, L1c data have been downloaded from the EUMETSAT Earth Observation data portal (https://eoportal.eumetsat.int,

last access: 16 July 2019) in NETCDF format. Data files also contain the co-located land mask and cloud fraction values, obtained from the Advanced Very High-Resolution Radiometer (AVHRR) measurements, also on board Metop.

### 2.2.2 MLS

The Microwave Limb Sounder (MLS) provides vertical profiles of several chemical components, by measuring the microwave thermal emission from the limb of Earth atmosphere (Waters et al., 2006). More than 2500 vertical profiles are observed daily,

including trace gases with a vertical resolution of approximately 3 km. Several studies benefited from MLS products, notably



the ozone profiles in assimilation experiments (Emili et al., 2014; Massart et al., 2009), thanks to its low bias in the stratosphere (<5%) (Froidevaux et al., 2008).

In our study, we use the ozone profiles retrieved from MLS (V4.2 Products) as independent data to validate our results. The data have been downloaded from the Goddard Earth Sciences Data and Information Services Center (GES DISC) web portal (https://disc.gsfc.nasa.gov).

### 2.2.3 OMI

The Ozone Monitoring Instrument (OMI) is a nadir-viewing, ultraviolet–visible (UV-VIS) spectrometer (Levelt et al., 2018). It provides complete global maps of total column ozone on a daily basis. The OMI ozone data record starts in October 2004, shortly after the launch of Aura (McPeters et al., 2015). The total column averaged over the month of the study (July 2010), resulting from the OMI-TOMS version 8 algorithm (Bhartia, 2002), is used here to validate the results of the assimilation experiments.

### 2.2.4 Ozonesondes

Ozonesondes are in situ instruments carried by a radiosonde transmitting continuously the measurements as the ascent in the atmosphere. The profiles of O3 are provided up to an altitude that often exceeds 30 km (Jiang et al., 2007) with a vertical resolution of 150-200 m. They have been used for several applications such as validating satellite products (Jiang et al., 2007). In our study, vertical profiles of ozone, collected and distributed by the Word Ozone Ultraviolet Radiation Data Centre (http://www.woudc.org, last access), are used to validate the model simulations.

### 2.3 Setup of the numerical experiments

The main purpose of this study is to estimate the IASI observation error covariance and verify its impact on the quality of the ozone assimilation results. Thus, we kept the same setup of (Emili et al., 2019) in terms of the period of the study, the model configuration, the choice of assimilated observations, and of the background error covariance matrix. The observation error covariance matrix will be discussed in the section of the results (Section 3).

The model was initialized with a zonal climatology and the spin-up time used is one month (June 2010). Then, our simulations were performed for the month of July 2010. The forecast error standard deviation was assumed to be proportional to the ozone concentration. In fact, Emili et al. (2019) have evaluated the standard deviation of the free model simulation against independent data (profiles from ozonesondes and MLS), and found out a small free forecast error in the stratosphere, larger error in the free troposphere and highest error close to the tropopause. The background standard deviation was, thus, taken equal to 2% above 50 hPa and 10 % below to mimic the validation's behavior. Similar choices were employed in (Massart et al., 2012; Peiro et al., 2018).





The background error covariance matrix is split into a diagonal matrix filled with the standard deviation and correlation matrix modeled using a diffusion operator. The correlation, characterized by the length-scale, spreads the assimilation increments in space. In this study, we keep the same configurations of horizontal and vertical length scales as in (Emili et al., 2019) .

The same preprocessing described in Emili et al. (2019) has been applied to our data before their use in the assimilation

system. In order to avoid any contamination from clouds, data were filtered using a cloud mask and only pixels with cloud fraction less than or equal to 1 % were kept. The cloud fraction values are obtained from the AVHRR measurements onboard Metop. Since the spatial resolution of MOCAGE is coarser than the pixel size, the number of ground pixels was reduced by thinning the data using a grid of 1° x 1° of resolution and only keeping the first pixel that falls in every two grid boxes. A dynamical rejection of observations - with a threshold of 12 % - based on the relative differences between simulated and measured

values with respect to simulated values was considered. Some channels affected by $H_2O$ absorption (1008-1019,1028-1030, 1064-1067,1072-1076,1089-1092 $cm^{-1}$) were removed. Pixels affected by aerosols are detected and then removed using the index based on V-shaped sand signature as discussed in Emili et al. (2019).

## 3   R estimation

### 3.1   Desroziers diagnostics

The observations used in the assimilation system could have a margin of error. We can identify two types of errors, systematic and random errors. The systematic error is ordinarily corrected before the data assimilation process. Nevertheless, this correction should not account for the model bias, which requires some independent data to prevent the drift of the analysis to unrealistic values. In our case, we are assimilating troposphere and stratosphere data, the missing of an anchor instrument in the top of stratosphere led us to assume that our observations are unbiased. This assumption, in fact, is adopted in most chemical

analyses (Emili et al., 2019; Massart et al., 2012). Random errors can arise from the measurements (e.g. instrumental error), forward model, representativeness error (e.g. difference between point measurements and model representation), or quality control error (e.g. error due to the cloud detection scheme missing some clouds within clear sky only assimilation). These types of errors should be accounted by the observation error covariances matrix $\mathbf{R}$. According to (Weston et al., 2013), the instrument noise could be assumed to be uncorrelated. However, the IASI measurements are apodized, which may introduce

correlations between neighboring channels, particularly in our case where we are assimilating a subset of adjacent channels. The radiative transfer may also introduce correlations between channels. The statistics of error from the instruments noise are known, while the characteristics of other sources of error are not yet well understood.

In this paper, we estimate the total error using the statistical approach introduced by Desroziers et al. (2006) .

$$\mathbf{R} = \mathbf{E}[(\mathbf{y} - \mathbf{H}(\mathbf{x}_a))(\mathbf{y} - \mathbf{H}(\mathbf{x}_b))]$$

Where $\mathbf{x}_a$ is the analysis state vector, $\mathbf{x}_b$ is the background state vector, $\mathbf{y}$ is the vector of observations and $\mathbf{H}$ is the observation operator that computes model counterpart in the observation space.





This method has been used to estimate observation errors and inter-channel error correlations (Stewart et al., 2009). It can potentially provide information on imperfectly known observation and background-error statistics with a nearly cost-free computation (Desroziers et al., 2006). However, this approach assumes that the $\mathbf{R}$ and $\mathbf{B}$ matrices used to produce the analysis are exactly correct, which may not be always the case. Furthermore, Desroziers diagnostics compute the total covariances,

more efforts are needed to understand and distinguish the sources of the error.

## 3.2  Error results

The Desroziers statistics were computed on the output of a 3D-Var experiment using a diagonal matrix $\mathbf{R}$ (with a standard deviation of 0.7 mWm$^{-2}$sr$^{-1}$cm as in (Emili et al., 2019). The diagnosed $\mathbf{R}$ could not be used directly in the assimilation system. In fact, the estimated matrix was asymmetric and not positive definite. This might result from the violation of one

of the assumptions of Desroziers statistics stipulating that the $\mathbf{B}$ and $\mathbf{R}$ matrices used to produce the analysis were correctly defined. Similar unrealistic features in the diagnosed covariance matrices were encountered in (Stewart et al., 2014; Weston et al., 2013) where an artificial inflation of observation errors was assumed. $\mathbf{R}$ needs to be a valid covariance matrix before being used in the 3D-Var assimilation system. Therefore, we first symmetrize the estimated matrix by taking the mean of the original and its transpose. Then we impose the negative eigenvalues to be equal to the smallest positive eigenvalue (Charles F.

Van Loan, 1996). An other method was tested here to recondition the estimated matrix, the one called *ridge regression* (Weston et al., 2013; Tabeart et al., 2020b) which consists of increasing all eigenvalues of $\mathbf{R}$ by the same amount. We favoured the first method since the standard deviation and the correlation values remain closer to the initially estimated quantities.

In order to evaluate the impact of starting from an experiment based on a diagonal $\mathbf{R}$ matrix, we performed a second 3D-Var experiment using the diagnosed and modified matrix. Then, we used these new forecasts and analyses to estimate again $\mathbf{R}$ with

the Desroziers diagnostics. The resulted standard deviation was significantly greater than the one estimated from the experiment that use a diagonal matrix (not shown). The same goes for the correlations (not shown). It should be noted that the re-estimated matrix was positive definite, unlike the first estimation where some unrealistic features were encountered. We have followed the same process to reestimate two other matrices (introduce, each time, the estimated matrix into the assimilation system in order to have analyses that we use to re-estimate a new matrix). The differences between the estimations in terms of standard

deviation and correlations became smaller as we reestimate the matrices (not shown), suggesting a sort of convergence of the estimation. All matrices and simulations discussed in this paper are based on the second estimation (which uses the analyses derived from the experiment using the first conditioned estimation). The reason for this choice will be rediscussed later (section 5.2).

Figure 1 presents the standard deviation diagnosed (second estimation) using the Desroziers approach (solid black line) and

that used in Emili et al. (2019) (dotted blue line). The latter was set equal to 0.7 mW m$^{-2}$ sr$^{-1}$cm for all channels, which is a common setting for most IASI O3 retrievals (Dufour et al., 2012). At first glance, we note that the standard deviation used in previous studies is highly underestimated for the SST sensitive channels and overestimated for some ozone sensitive channels (around 1040 and 1050 cm $^{-1}$ ). The diagnosed standard deviation increases to reach 2 mW m$^{-2}$ sr$^{-1}$ cm for SST sensitive channels (the first and the last twenty channels of the band (980-1000 cm$^{-1}$ and 1080-1100 cm$^{-1}$ ) and the channels between



1040 and 1045 cm$^{-1}$ ) and varies from 0.2 to 1.4 mW m$^{-2}$ sr$^{-1}$ cm for the ozone sensitive channels. The radiance values for the observations are greater for the SST channels than those of the ozone. The same goes for the corresponding background and the analysis values. Since these diagnostics are based on observation, background and analysis residuals, a larger standard deviation for the SST channels than for ozone channels might be expected. We remarked that the estimated standard deviation

is proportional to the radiance value (either the observation, the background, or the analysis value), which gives a relative standard deviation of about 2.5 % of the average radiances values for the entire spectral window (not shown).

The IASI instrumental error is provided by the CNES (Centre National d'Etudes Spatiales ), taking into account different known effects such as flight homogeneity and apodization effect (Le Barbier Laura, personal communication, September 18, 2019). The instrumental error covariance matrix is computed as described in (Serio et al., 2020). This error remains smaller

(maximum of 1.06e$^{-6}$ mW m$^{-2}$ sr$^{-1}$cm) than that used in the previous studies (0.7 mW m$^{-2}$ sr$^{-1}$cm). Then, the important estimated standard deviation noticed in our estimation might be due to the radiative transfer inputs error.

To investigate the off-diagonal part of $\mathbf{R}$ we present the correlation matrix in figure 2. The results show high correlations between the majority of the channels (larger than 0.4 ). In particular, a very high correlation is observed among SST sensitive channels (around 0.9 to 1). The regions of, relatively, lower correlation (around 0.4 to 0.7) represent the ozone channels

correlations and cross correlation between ozone and SST sensitive channels.

The high correlation found here was expected since previous studies have highlighted the same behaviour in this spectral region (Bormann et al., 2010; Stewart et al., 2014; Bormann et al., 2016). In fact, the use of the same radiative transfer model for all channels may introduce similar errors among these channels.

The diagnostic discussed above is based on a global estimation, without any distinction between the type of the surface (land

or sea) nor the time of the observation (day or night). Since the emissivity varies according to the type of the surface, and the skin temperature is strongly driven by the sun radiation, we evaluated $\mathbf{R}$ taking these differences into account.

Figure 3 shows the difference between the standard deviation estimated using data localized over the sea and land separately. The error over land reveals large values for the SST sensitive channels in comparison with that estimated over the sea which, in turn, reproduces a slightly different error in comparison with the global estimation. The two surfaces introduce also a slightly

different error regarding the ozone band. Since the number of observations over the sea is greater than over the land (almost 65 %), the global estimation is dragged down to be close to the sea estimation. To explain the large difference in the region of SST sensitivity channels over the land, the surface properties need to be discussed.

The surface emissivity varies with vegetation, soil moisture, composition, and roughness (Nerry et al., 1988) with values between 0.65 and 1 in the thermal infrared range. Low values are found in deserts and high values over dense vegetation and

water surface (Capelle et al., 2012). The variability of the soil type over land in comparison with the sea that is homogenous can lead to a larger standard deviation over land than over sea. The second parameter that impacts the electromagnetic spectrum measured by the infrared sensor is the surface temperature. Its variability is larger over land than over sea, which also contributes to the difference of the standard deviation between land and sea. On the other hand, these differences are larger for the SST channels than for the ozone channels. This might be explained by the direct link between the radiance signal and the





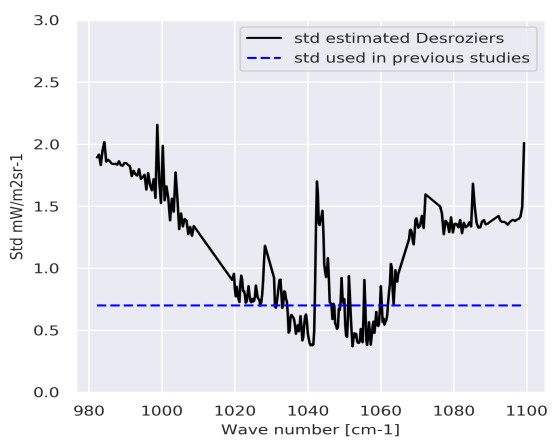

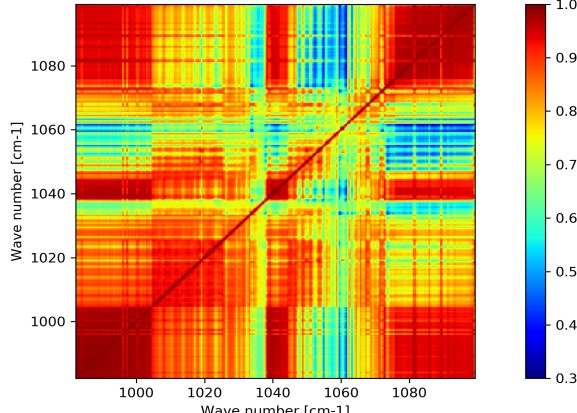

**Figure 1.** Standard deviation estimated using Desroziers diagnostics (back solid line) and that used in the previous studies (dotted blue line).

**Figure 2.** Correlation matrix estimated using the statistics of Desroziers.

surface signal in the SST bands especially since the observation and the estimated standard deviation were found proportional (mentioned above).

The same behaviour as the global estimation is reproduced when the statistics were performed from the data measured separately from the day and from the night. Figure 4 shows small differences in the SST channels sensitivity band (between
5  1080 and 1100 $cm^{-1}$ ), whereas the curves take approximately the same values for the rest of the channels.

The data assimilation process uses the full error covariance matrix in order to minimize the cost function. Hence, we discuss now the differences between the correlation matrices taking the time (day/night) and the surface type (sea/land) of the observations into account. Figure 5 shows the correlation matrix estimated using all observations (a), only pixels over sea (b), only pixels over land (c), the difference (in %) between the global and sea matrix (divided by the global matrix) (d), and the
10  difference (in %) between global and land matrix (divided by the global matrix) (e). According to the figures (a), (b), and (c), the differences between the three matrices differences may appear small. However, by looking at the figures (d) and (e), the difference between the correlations estimated using all observations and pixels over the sea surface varies between 0 % and 40 % for the majority of the channels with values that can reach 60%. These differences are located around 1035 and 1060 $cm^{-1}$ which correspond to the regions of low correlations.
15  The same comparison was held separating observations from the day and from the night: significantly smaller differences compared to the land and sea separation were noticed (not shown).





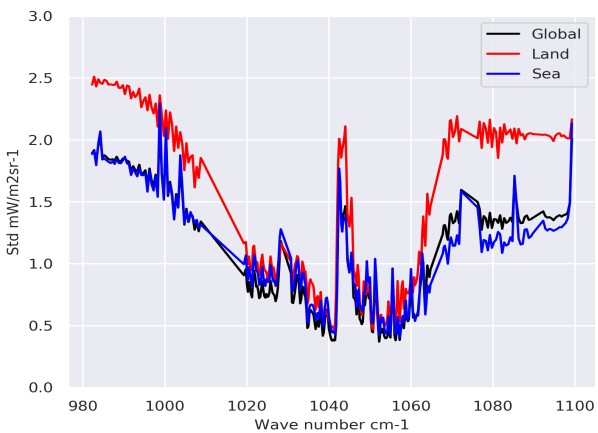
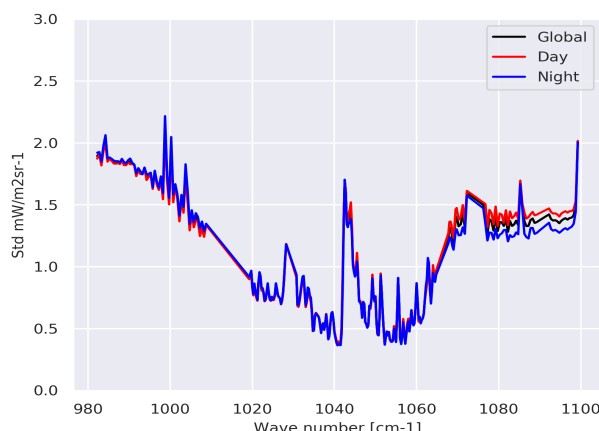

**Figure 3.** Standard deviation estimated using Desroziers diagnostics according to the type of the surface (sea, land and global).

**Figure 4.** Standard deviation estimated using Desroziers diagnostics according to the time of the observation (day, night and global) .

In conclusion, the variability in terms of correlations is more pronounced when the surface type is considered than in the case of the observation time. For the latter, the smaller differences found for the standard deviation (Fig. 4) are also found for correlations. For the follow-up assimilation experiments, we adopted the global estimation and neglected the effect of the time and surface of the observations. The rationale for this choice will be given during the discussion of the validation results
5    (section 5.2).





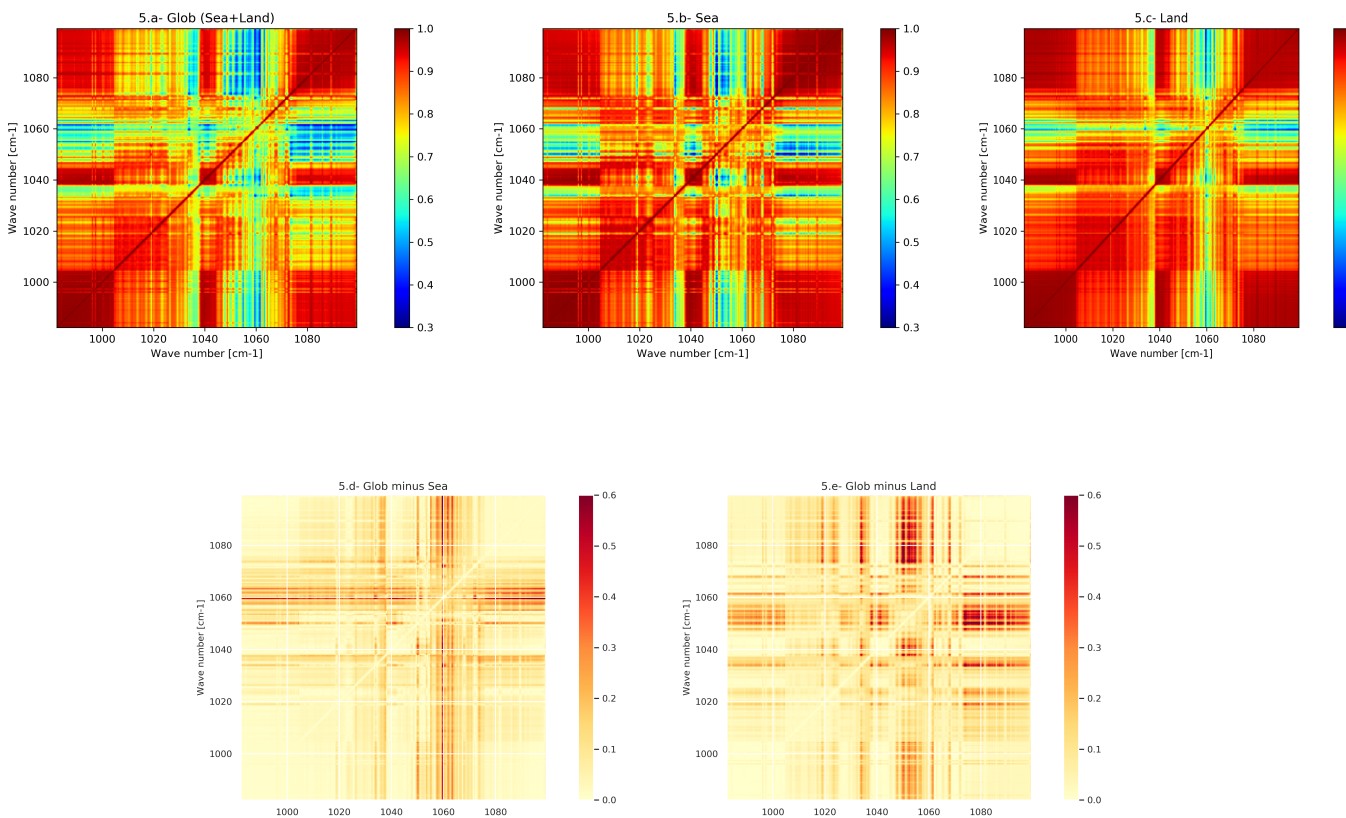

**Figure 5. (a)** Correlation matrix estimated using all observations, **(b)** using pixels observed over the sea, **(c)** using pixels observed over the land, **(d)** difference (in %) between global and sea matrix (divided by the global matrix), **(e)** and the difference (in %) between global and land matrix (divided by the global matrix)

# 4 Assimilation results

## 4.1 Ozone fields

In this section, we discuss the impact of the observation error covariance estimated previously on the ozone analysis. To this end, three experiments for the month of July 2010 were carried out:

5    i). a model run without data assimilation called hereafter the free run (or Control), and noted in the rest of this paper ControlExp.

ii). A 3D-Var assimilation of IASI radiances using a diagonal observation error covariance matrix (as in Emili et al. (2019)). It will be referred here by RefExp.



iii). A 3D-Var assimilation of IASI radiances using a full matrix estimated with the Desroziers diagnostic noted hereafter by RfullExp.

The first experiment (ControlExp) was run to evaluate the benefit of the assimilation experiments and to quantify the improvements of each of the two analyses when they are validated against independent data. The same setup of Emili et al. (2019) was adopted for RefExp, which was taken as a reference to characterize the impact of accounting for the estimated $\mathbf{R}$ in the third simulation (RfullExp).

Figure 6 shows the difference between the zonal average of the analysis from the two assimilation experiments in units of parts per billion volume (ppbv). The zonal values were averaged over the month of the study before performing the difference. The impact of the estimated $\mathbf{R}$ varies with latitude. It varies also with the height, adding or reducing the amount of ozone. Overall, the estimated $\mathbf{R}$ reduces the amount of ozone in the high latitudes of the free troposphere and the tropical high stratosphere, whereas the amount is increased in the vicinity of the lower stratosphere. The maximum reduction of ozone is larger than the amount added. The amount of ozone reduction reaches 600 ppbv, whereas the increase does not exceed 300 ppbv. In high northern latitudes (30°N-90°N), a significant addition is found (300 ppbv) covering almost the whole stratosphere, in opposition to the other latitudes where the difference changes sign with altitude. On the other hand, an important reduction of ozone is observed in the tropics at 20 hPa (more than 600 ppbv). To better understand the impact of the estimated $\mathbf{R}$ we validate the results with independent data in the section of validation (section 5).

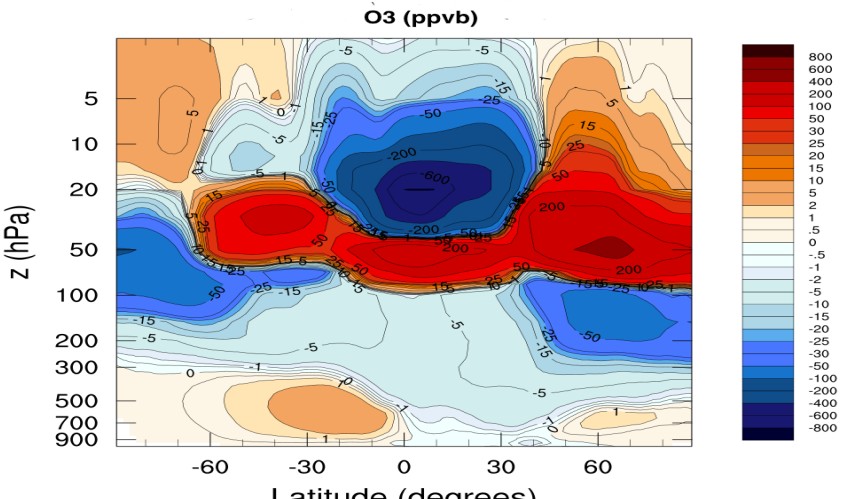

**Figure 6.** The difference between the zonal average of the analysis (in ppbv) from the two assimilation experiments, averaged over the month of the study (nonlinear colormap).





The emphasis, up to now, has been on the impact of the estimated observation error on the ozone analysis. We discuss in the following subsection the impact on the skin surface temperature analysis.

## 4.2 Surface skin temperature

The assimilated spectra include both ozone and surface skin temperature sensitive channels. The IFS skin temperature was taken as a background in the assimilation process. We have computed the difference between the SST analysis and the background at the end of each assimilation experiment (RefExp and RfullExp). Figure 7.a) shows the difference between the analysis of the SST given by RefExp and the IFS SST forecast whereas, figure 7.b) shows the difference between the analysis of the SST given by RfullExp and the IFS SST forecast. In terms of geographical distribution, we notice that the differences are smaller through the tropics and mid-latitudes, especially over sea, when the estimated $\mathbf{R}$ was adopted. Looking at the average values, RefExp decreases the surface skin temperature by about 0.7°C with respect to the background. The introduction of the estimated $\mathbf{R}$ decreases the difference between the SST analysis and that of IFS to almost -0.3°C instead of -0.7 °C. The standard deviation was also reduced from 1.75 °C. to 1.43 °C. Thus, the use of the estimated $\mathbf{R}$ lets the SST analysis stay closer to the IFS forecasts which are reliable data.

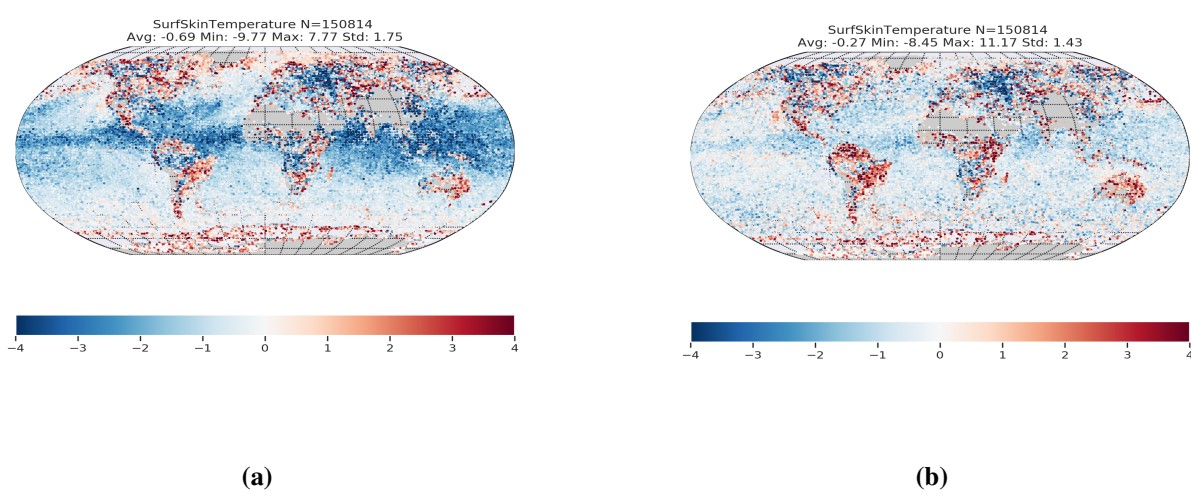

**(a)**                                                    **(b)**

**Figure 7.** Difference ( in °C ) between the IFS SST forecast and the analysis of the SST given by RefExp (with a diagonal matrix) **(a)**, and that given by RfullExp (with a correlated matrix) **(b)**.

## 4.3 Computational cost

In our assimilation setup, the cost function is minimized hourly. For each window, the minimizer needs to converge after a certain number of iterations. The cost of each iteration is dominated by the cost of the radiative transfer operators (tangent linear, the adjoint model) and of the background error covariance operators. When the observation error was assumed to be





uncorrelated (RefExp), the number of iterations needed for each hourly cycle is significantly higher than when the estimated observation error covariance matrix is used. In fact, the introduction of the estimated $\mathbf{R}$ reduces the number of iterations from 150 (a fixed value to stop iterations if the convergence criteria were not achieved to save computational time) to 90 iterations in average. This means that the CPU time is reduced by more than 150% for each assimilation cycle. This improvement is

highly valuable when the study is extended to longer periods. This reduction is due to the fact that the diagonal matrix is pulling much closer to the observations than the correlated matrix, which makes it harder to find a solution resulting in slower convergence. This increase of the convergence speed was encountered in the Met Office 1D-Var system (Tabeart et al., 2020a) where a correlated observation matrix was introduced in the system. Furthermore, in Tabeart et al. (2018) the matrix $\mathbf{R}$ and the observation error variance appear in the expression of the condition number of the Hessian of the variational assimilation

problem, indicating that these terms are important for convergence of the minimization function. To understand which one between correlations and variance has a higher impact on the convergence speed, we proceeded as follows: we considered an estimation that did not lead to a convergence (the first estimation) of the minimizer, we kept its correlations and replace its standard deviation with that of an estimation that leads to a convergence (the second estimation). The minimizer, consequently, converges allowing us to conclude that the variance has a higher impact on the convergence speed.

## 15   5   Validation of O3 analyses

### 5.1   Total column

Figure 8 shows the difference of the ozone total column (in Dobson Unit (DU)) provided by OMI and that of the RefExp (a) and that of RfullExp (b). At first sight, we note smaller differences over the tropics between the OMI total column and the total column given by RfullExp in comparison with that given by the RefExp. This behaviour was expected since an important

reduction of the amount of ozone was observed in these regions (see figure 6). In the high northern latitudes, the differences were slightly increased when the estimated matrix was adopted. This is a consequence of the increase in the amount of ozone encountered in these regions the stratosphere, compared to the amount reduced in the same region in the troposphere (figure 6). On the other hand, the global mean and the standard deviation of these differences are lower in the case of using the new estimated matrix.

Hence, we conclude that the estimated matrix $\mathbf{R}$ has slightly improved the results in terms of ozone total columns.



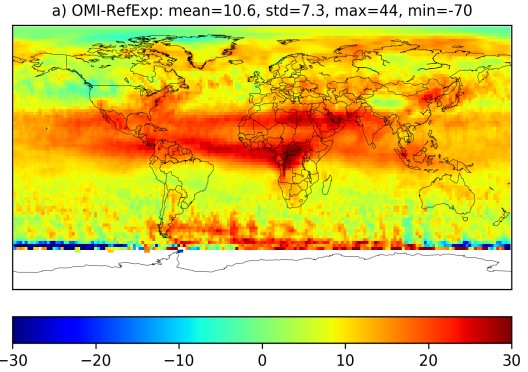

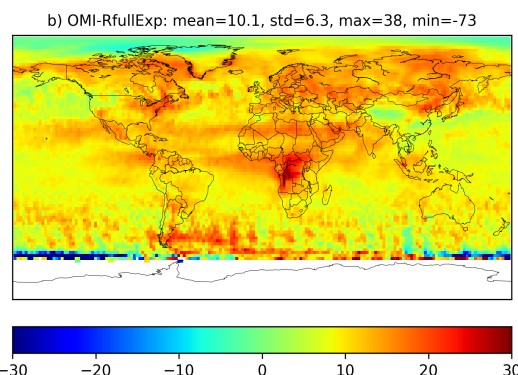

**Figure 8. (a)** Difference of the ozone total column (DU) provided by OMI and that of the assimilation experiment RefExp,**(b)** and that of RfullExp, averaged over the month of the study.

## 5.2 Vertical validation

In this section, we validate the two simulations against radiosoundings and MLS data. We use the root mean square error (RMSE) as the main statistical indicator to quantify the accuracy of the experiments.

We compute the relative (to the control simulation) difference of RMSE with respect to radiosoundings and MLS averages globally and for five different latitude bands. The difference is computed by subtracting the RMSE of each experiment from that of the control simulation. Negative values indicate an improvement of the O3 profiles. It should be noted that the representativeness of the statistics given by the MLS in the stratosphere is better than that of the radiosoundings because the number of profiles provided by MLS is much higher compared to the radiosoundings ones. Consequently, higher confidence is given to the validation using the MLS data in the stratosphere.

Figure 9 reports the RMSE differences with respect to the radiosoundings. Considering the global RMSE (ALL), we notice that the experiment with the estimated matrix improves the results above 150 hPa, around 400 hPa and near the surface.





However, it also reduces the improvement from 30% (the case of using a diagonal matrix) to 15 % in the vicinity of UTLS (100-300 hPa).

The problem of increasing the error in the stratosphere reported in Emili et al. (2019), is especially severe in the tropics (30S-30N). The use of the estimated **R** has substantially enhanced the results in this latitude bands bringing the error from

+15% to -2%. Apart from the vicinity of 50 hPa and 400 hPa, the results, in the tropics, were improved over the entire vertical profile. Regarding other latitude bands, almost the same feature of the global validation is found in the north hemisphere. The two experiments show almost the same behaviour in the southern latitudes, with a slight improvement for RfullExp in the southern high latitudes (60°S-90°S).

The MLS validation in figure 10 shows almost the same behaviour reported by radiosoundings validation in the tropical

stratosphere, where the reduction of error is remarkable. In the other latitude bands, MLS reports a similar behaviour of the two experiments, with some small differences in northern hemisphere.

All things considered, the introduction of the estimated **R** has globally improved the O3 profiles in the stratosphere and in the free troposphere, especially in the tropics. In spite of its degradation in the vicinity of UTLS, the improvement remains always advantageous with respect to the control run.

The matrix used for this study (see Sec. 3.2) will be now discussed in this section since the decision was also based on the outcome of the assimilation experiments presented in this section. We performed sequentially three assimilation experiments using the first, second and the third estimation of **R** (Sec. 3.2). The results of validation against radiosoundings and MLS showed small differences (not shown). Therefore, to avoid the initial impact of using a diagonal matrix we have adopted the second estimation (which uses the analyses derived from the experiment using the first conditioned estimation). In an

operational framework, we may estimate the matrix daily (weekly or monthly if the period of the study is considerably long) using the analyses of the previous day (using the analysis of the previous week or month respectively ). In other words, we may use a diagonal matrix to produce analyses for the first day or spin-up period, these analyses will be used to estimate the matrix that will be used for the second day, and so on throughout the period of the study.

We have also discussed the type (sea/land) and the time (day/night) of the observations while estimating the matrices. To

check the impact of these differences on the assimilation results, we ran an additional assimilation experiment using the matrix estimated considering the type of the surface of each observation (since the differences were more important than if the time of the observation was considered). Only slight differences among the results have been noticed (not shown). Thus, for simplicity, it seems reasonable to adopt the global estimation of the matrix and neglect the effect of the time and the type of the surface of the observations.

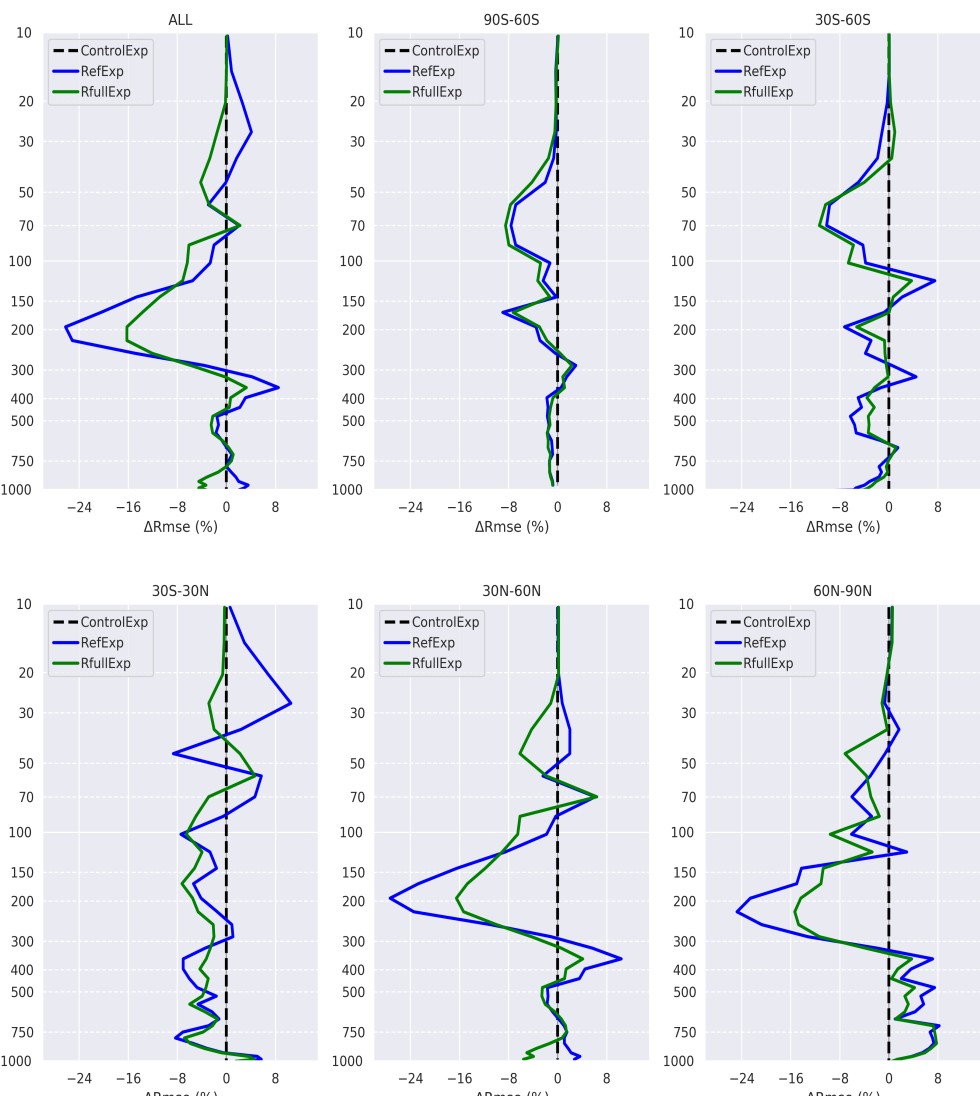

**Figure 9.** Relative difference of the RMSE with respect to radiosoundings for the RFullExp (green) and RefExp (blue). The relative difference of the RMSE was computed by subtracting the RMSE of the controlExp from the RMSE of the analysis of each experiment, divided by the average profile of radiosoundings.





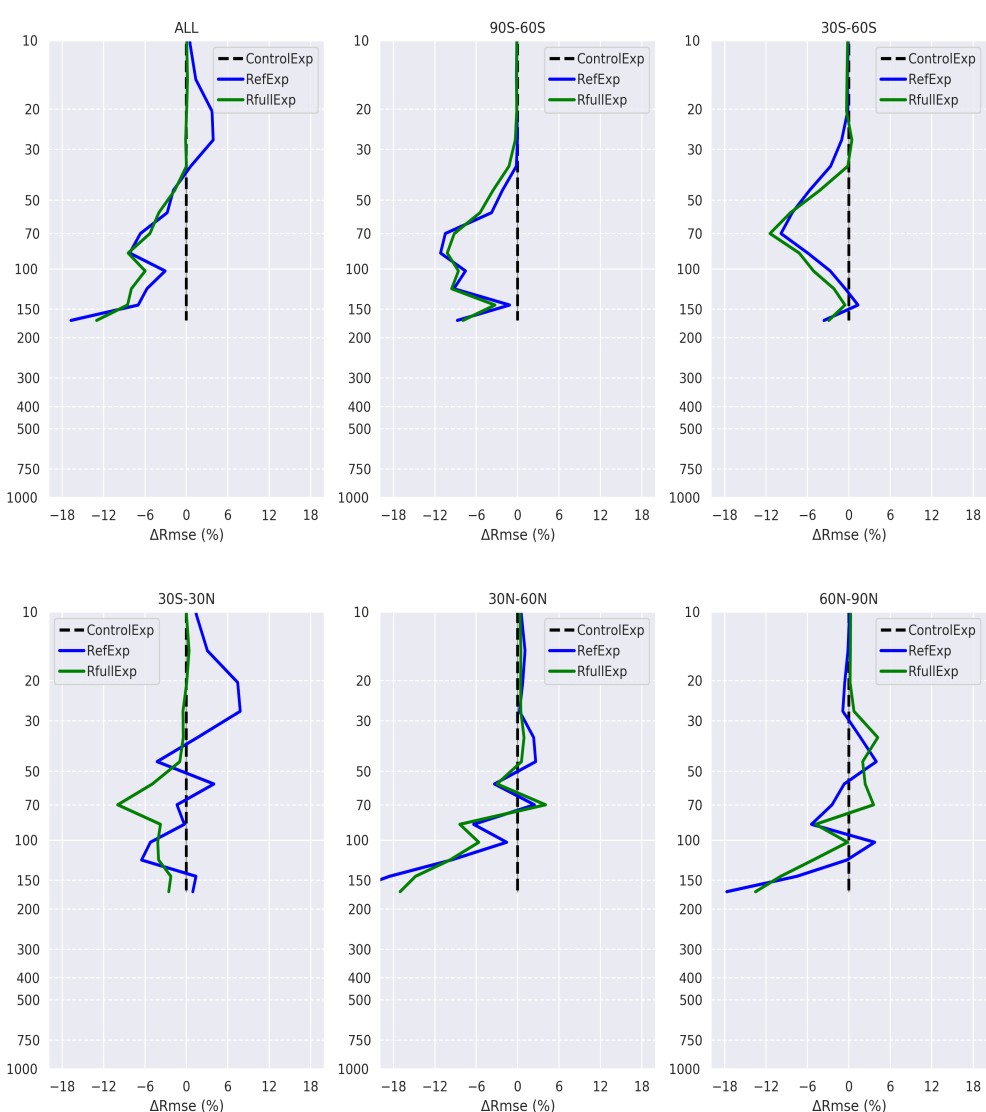

**Figure 10.** Relative difference of the RMSE with respect to MLS profiles for the RFullExp (green) and RefExp (blue). The relative difference of the RMSE was computed by subtracting the RMSE of the controlExp from the RMSE of the analysis of each experiment, divided by the average profile of MLS.





## 6  Conclusions

The correct specification of the observation error becomes a critical issue to assimilate efficiently the increasing amount of satellite data available in the recent years. We have estimated the observation errors and their inter-channel correlations for clear sky radiances from IASI ozone sensitive channels. We have evaluated, then, the impact of the estimated $\mathbf{R}$ on the SST and ozone analysis within our 3D-Var assimilation system. The outcome has been compared with an assimilation experiment where the observation error covariance matrix was assumed to be diagonal and the standard deviation assigned empirically like in previous studies. The results of the experiments were, then, validated against independent data: OMI, MLS and ozonesondes.

The Desroziers diagnostics were adopted here to estimate $\mathbf{R}$. The diagnostics used the analyses derived from a variational data assimilation experiment. The results have shown high correlations between the majority of the IASI spectral channels, particularly among the SST sensitive channels.

The impact of the surface type (land/sea) and the time of the observation (day/night) on the estimation was evaluated. Significant differences were noticed in terms of standard deviation when the surface type was considered. However, slight differences in the results have been noticed when the matrix estimated considering the time and the type of the surface of the observation was used in the assimilation experiment. A global estimation of the matrix could finally be adopted.

Significant differences between the results of the experiments were encountered. The introduction of the estimated $\mathbf{R}$ reduces the amount of ozone in the free troposphere and in the high tropical stratosphere, whereas ozone is added in the vicinity of the lower stratosphere. The total column also has shown differences on average and in terms of geographical distribution. A validation against OMI has shown that the results were closer to the observations when the estimated matrix was adopted.

The validation against MLS and ozonesondes showed that the introduction of the estimated $\mathbf{R}$ has globally improved the results in the stratosphere and in the free stratosphere especially in the tropics. In spite of a slight reduction in accuracy in the vicinity of UTLS, the improvement remains always advantageous with respect to the reference assimilation. Concerning the computational cost, the introduction of the estimated $\mathbf{R}$ significantly reduces the number of iterations needed for the minimizer to converge.

In summary, accounting for an estimated $\mathbf{R}$ improves significantly the ozone assimilation results. Furthermore, this approach might be adopted in the assimilation of other chemical species and also in a chemical retrieval process framework.

In this study, the estimation was computed without taking into account any distinction of the error sources and assuming that the observation error was unbiased. More efforts will be needed to tackle these points. It should also note that we kept the same experiment setup of Emili et al. (2019) in order to be able to quantify exclusively the impact of the $\mathbf{R}$. The background error matrix was still defined using a relatively simple and empirical method. Further research might be needed to perform a better estimation of the background error. A new channel selection might also be performed to reduce the computational cost and the information redundancy of the IASI spectrum. On the other hand, all the experiments are performed in the context where aerosols are neglected and over one month. Including modelled aerosols within the radiative transfer may bring some improvements to the analyses. These aspects will be covered in future research.



*Competing interests.* The authors declare that they have no conflict of interest.

*Acknowledgements.* We acknowledge EUMETSAT for providing IASI L1C data, WOUDC for providing ozonesondes data and the NASA
Jet Propulsion Laboratory for the availability of Aura MLS Level 2 $O_3$. We also thanks the MOCAGE team at Météo-France for providing
the chemical transport model, the RTTOV team for the radiative transfer model, and Gabriel Jonville for the help on technical developments
5   of the assimilation code. This work has been possible thanks to the financial support from the Région Occitanie.





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
