# Peer review of "Estimation of the error covariance matrix for IASI radiances and its impact on the assimilation of ozone in a chemistry transport model"

_Atmospheric Measurement Techniques, 2020_

## Referee Comment (RC1) · Anonymous Referee #1 · 23 Sep 2020

Overview

This paper proposes a method to estimate the error covariances of ozone-sensitive Infrared Atmospheric Sounding Interferometer (IASI) channels and to evaluate their impact on the ozone analyses in the MOdèle de Chime Atmospherique à Grande Echelle (MOCAGE) chemistry transport model. A set of 280 channels between 980 and 1100 cm-1 is used for this study. The author chose to diagnose his observation-error covariance matrix (R) using the method of Desroziers et al. 2005, which allows the estimation of inter-channel error covariances. Different 3D-Var data assimilation experiments are

performed to provide ozone analyses that have been compared to independent data (ozone-sondes, the Microwave Limb Sounder (MLS) and the Ozone Monitoring Instrument (OMI)).

General Comments

Overall, the paper is well structured. The improvement in the quality of the figures is noticeable. This study deals with an interesting and in progress subject, which is data assimilation in chemistry transport models. The comparisons between observation-errors according to surface types and between day and night are interesting. However, a guideline is missing. What is the main objective of this paper? To diagnose an R-matrix or to improve ozone analyses using a diagnosed R-matrix?

Then, I understand that it is long and expensive to carry out these experiments but I wonder about the significance of an experiment of one month. It would have been beneficial to continue these experiments over 2 months, as well as over two distinct periods (summer and winter). In addition, this paper lacks a discussion about the bias correction that may be needed for ozone-sensitive channels. A comparison with the work of (Han and McNally, 2010) would have been relevant.

Finally, it is useful and important to refer to previous studies, however the constant reference to the work of (Emili et al. 2019) is over-exploited. This paper would benefit from providing all the technical information required for a good understanding of the characteristics of the experiments. I will provide you some specific comments on this in the following.

Specific Comments

Title: I think it is important to specify in the title, that this work is carried out in a chemistry transport model.

P1, L6: (...between 980 and 1100 cm-1) I suggest adding that this spectral range includes ozone-sensitive channels and atmospheric window channels.

P1, L11: (The computational cost...) This sentence is useless without explanation. I suggest you delete it or add a short comment.

P2, L30: (...impact on analysis accuracy.) Specify that this is the impact on the ozone analysis.

P3, L2: There are more recent studies on the same subject that you can reference: (Weston et al. 2014, Borman et al. 2016, Tabeart et al. 2020, Coopmann et al. 2020)

P3, L29: (...the radiative transfer model RTTOV) Most recent reference to the work of (Saunders et al. 2018).

P3, L31: (...Starting from an atmospheric...) Specify that RTTOV requires a vertical temperature and humidity profile.

P4, L6: What about other chemical variables (CO2, CH4, CO, N2O, SO2)? Do you use reference profiles? Which coefficient file do you take into account?

P3, L17: Indeed, the observation-error variances and observation-error covariances plays a fundamental role in the data assimilation process. In addition, background-errors are also very important in this process. For the purpose of consistency, It is required, at least, to show the background-error variances or background-error error standard deviation, as well as, the background-error correlations matrix.

P3, L19: (as a percentage of the observation values.) What does this percentage look like?

P3, L22: Are there other variables included in the control vector?

P5, Table 1: Can you provide more information about the ozone background?

P5, L15: (...co-located land mask...) Wouldn't it be the "Land Sea Mask" instead?

P5, L16: In this case, from which satellite platform are IASI observations extracted? MetopA, B, C?

P6, L20: Another reference to (Emili et al. 2019)... It would be very useful to summarize the configuration of the experiments in a table.

P6, L27: Can you compare these ozone background-error standard deviation with other values used in recent research?

P7, L11: On what criteria were these channels identified as sensitive to water vapour?

P8, L18 to L28: This paragraph is complicated to follow and it is a pity because it is important for the next step. I suggest you summarize the different configurations in a table.

P10, Figure 2: Correlation matrices can vary between -1.0 and 1.0. I expected to see negative correlations between some channels in the atmospheric window and some ozone-sensitive channels. Why not represent the matrix between -1.0 and 1.0, centered on white at zero?

P12, Figure 5: Same remark as above about the color scale.

P12, L8: The naming of the experiments is not appropriate because one could confuse Control and Reference. I would suggest RdiagExp instead of RefExp.

P14, L4: It would be useful to explain the physical link between skin temperature and ozone in the assimilation of infrared observations. Is there any consideration of inter-variable background-error correlations between O3 and Tskin?

P14, Figure 7: There is also increase in difference on land using RfullExp, mainly in Africa and South America. This can be related to the differences in observation-errors depending on the surface... In addition, there are too many pixels on the map. It would be interesting to average by box in order to better exploit the information provided by this Figure.

P17, L15 to L17: This paragraph is not clear... Where does this third estimate come from? It does not seem to me to have seen any explanation for it before. If this is the

case, it is not explicit and needs to be clarified.

Conclusions: I find that there is a lack of discussion about:

IASI channels between 1000 and 1070 cm-1 are mainly sensitive to ozone above 100 hPa, which poses the challenge of using other observations for a complete analysis of ozone over the entire atmospheric column...

Similarly, the high sensitivity of the ozone channels raises the problem of the amount of information remaining after a cloud detection...

Finally, work on background-errors is significant for the distribution of ozone increments...

Technical Corrections

P1, L4: (Modèle de Chimie Atmosphérique à Grande Échelle)

Throughout the paper: I suggest (Chemistry Transport Model) instead of (Chemical transport model)

P1, L5: (. . .already adopted in numerical weather prediction centers) This is not the case for all centers, (. . .already adopted in some numerical weather prediction centers)

Throughout the paper: Beware of the systematic use of (Furthermore). Vary the adverbs.

P2, L8: (. . .to construct a realistic picture of the...) The term (picture) is not appropriate, I suggest changing the word.

P2, L22: (...uncorrelated, some Numerical Weather Prediction...)

P2, L31: (. . .evaluate their impact on the ozone analysis accuracy)

P3, L19: (. . .MOCAGE is fed with forced by meteorological...)

P5, L5: (. . .the polar-orbiting satellite Metop-A, B and C launched...)

P6, L24: (The ozone forecast-error standard deviation...)

P6, L27: (The ozone background-error standard deviation...)

P7, L1: (The ozone background-error covariance matrix. . .)

P9, L12: (. . .we present the diagnosed correlation matrix...)

Throughout the paper: Be careful to capitalize the words (Figure)

P14, L16: (. . .to converge after a certain number of iterations)

Throughout the paper: Write rather with dashes (observation-errors, background-errors , ozone-sensitive,...)

P15, L22: (. . .encountered in these regions in the stratosphere...)

Please also note the supplement to this comment:
https://amt.copernicus.org/preprints/amt-2020-179/amt-2020-179-RC1-supplement.pdf

---

## Referee Comment (RC2) · Anonymous Referee #2 · 28 Sep 2020

This manuscript describes the estimation of errors and inter-channel error correlations in ozone and surface sensitive IASI observations. The estimated covariance matrix is then used when assimilating IASI radiances in the MOdèle de Chime Atmospherique à Grande Echelle chemical transport model. Accounting for correlated error in chemical transport models is a new and interesting area of research and the authors show benefit from using the new IASI error covariance matrix. There are some issues that need to be addressed, which I have outlined below.

General comments:

The authors present IASI error correlations for different surface types and for land/night.

I thought these results were interesting, but these matrices were not used in any of the assimilation experiments presented. I would welcome a discussion on why the separate treatment of the land/sea covariance matrices did not yield significant results. Otherwise I think the land/sea and day/night results could be cut.

The correlation matrices that are plotted show very strong inter-channel error correlations and almost look singular. What are the final condition numbers of the reconditioned covariance matrices? It seems that reconditioning was only used to correct negative eigenvalues, which could result in a nearly singular matrix. In fact I was surprised that the use of these matrices lead to a faster convergence of the minimization algorithm. What is the minimization algorithm, and does it include a preconditioner that depends on R?

Radiance bias correction is important in the assimilation of IASI channels that are sensitive to the surface and ozone. Was bias correction used and if so, what method? If not, were significant biases observed in the IASI observations?

The assimilation experiments were run for a 1 month period. Is this long enough to quantify the significance?

Are SBUV observations available for assimilation or validation of results?

The quality of English language should be improved before accepting this manuscript.

There are several incorrectly cited works and the references need to be carefully checked.

Specific comments:

P2 L4 and P5 L5, IASI is on Metop-A, B and C. But only Metop-A was available during the period of this study.

P2 L16 Other references exist that discuss sources of IR error and error correlations. Representivity errors can also contribute to inter-channel error correlations.

P2 L23 Campbell et al 2017 is another study to cite here.

P2 L31 Is the main objective to study the impact on ozone analysis accuracy? Also I think it is worth mentioning here that this is within the framework of a CTM.

P3 L1 There are numerous other studies that could be cited here.

P4 L22 Does the control vector include any other variables?

P5 L16. A brief discussion of channel selection is warranted. The abstract mentions that 280 channels are used, but this is worth restating here.

P6 L21 What observations are being assimilated? Assimilating observations from OMI, SBUV or ozonesondes could help anchor the bias correction of IASI ozone channels.

P7 L18 Change "missing" to "absence." Is there a better justification for this assumption?

P8 L1 There are many other references that should be cited in addition to Stewart et al, 2009. In addition, there have been a few theoretical studies on the Desroziers method that could be cited here.

P8 L8 How many days of data were used in the computation? How many days of data were used for the re-estimations?

P8 L9 The term "positive definite" is typically not used when discussing asymmetric matrices. A symmetric matrix is positive definite if and only if all of its eigenvalues are positive. I suggest that you instead discuss the eigenvalues after the matrix is made symmetric.

P8 L14 Citing Van Loan seems out of place here. Is there a specific page number? This again is an incorrect citation, Gene Golub was another author of this textbook. I believe this method was used in Weston et al 2014 and in Tabeart et al, 2020b, so these might be better references.

P8 L22 Positive definite- was the re-estimated matrix symmetric?

Figure 1- Please state in the caption what the "previous studies" are.

Figure 2, 5- I do not trust the tick labels on the x and y axes. In Figure 1, there seems to be a gap in channels between 1010 and 1020 cm-1. If the ticks on Figure 2 are linear then the plotting program might be interpolating the covariances between 1010-1020 cm-1, or the tick labels could be wrong. I know that Matlab for example does not label ticks correctly by default when making matrix plots like these. Also, in the caption, "statistics of Desroziers" would read better as "Desroziers method" or "Desroziers diagnostic"

Figure 5- Anti-correlations likely exist over land. I suggest changing the colorbar scale to include negative values.

P13 L7 Suggest to specify in this sentence that this is an ozone analysis.

P13 L14 Why is the reduction important?

P15 L10-14. I do not understand this discussion. What is the "estimation?" Why did the minimization fail to converge when using the "first estimation" but not the "second estimation?"

P15 L20 Why does a reduction of ozone in the analysis imply a better fit to OMI?

Figures 9 and 10, in the captions it would be helpful to state that negative values indicate an improvement in fit relative to ControlExp. Is it possible to remove the empty space in Figure 10? What is meant by "divided by the average profile of radiosoundings?"

P20 L16 Specify that ozone is added/reduced in the analysis.

P20 L17 "The total column also…" This sentence can be omitted. Isn't the total column result the validation against OMI mentioned in the next sentence?

Technical comments:

The work from Desroziers is not cited correctly. It is from 2005, not 2006. Weston et al. is cited incorrectly as well, this paper is from 2014.

Throughout the article, ozone is written as O3, when it should be $O_3$ with the 3 in the subscript.

P1 L22 Change "Remote sounding from satellites is" to "Remote soundings from satellites are"

P2 L1 Change "monitoring atmospheric gases, a large" to "monitoring of atmospheric gases. A large"

P2 L10 Should "Recent studies" be "A recent study" instead?

P2 L26 Remove (Weston et al 2013) from this sentence. All of the studies mentioned above show this result.

P2 L29 "(increase of the errors...)" The errors of what?

P3 L3 and other places. This should read "Desroziers method", not "Desroziers statistics"

P3 L7 Why not mention Section 5 in this paragraph?

P4 L10 "aerosols" should be singular.

P4 L18 What data file do you mean? I don't see one defined.

P4 L23 "evaluate the impact of the estimated observation error" and the error covariances, correct?

P4 L26 Change "reminded" to "given"

P6 L13 Change this sentence to "...carried by a radiosonde continuously transmitting measurements as it ascends."

P6 L26 Change "found out" to "found"

P7 L26 Change "radiative transfer may" to "radiative transfer model may" and "statistics of error from the instruments" to "error statistics from instrument"

P7 L29 The second term in the expected value should be a vector transpose.

P8 L7 "(with a standard..." there is no closing parenthesis

P8 L12 Change "assumed" to "applied"

P8 L15 Change "An other" to "Another"

P8 L 20 Change "The resulted standard deviation was greater than the one" to "The resulting standard deviations were greater than those"

P9 L 22 Figure 3 shows standard deviations, not differences.

P14 L1-2 These two sentences are unnecessary

P15 L25 This sentence should be a part of the previous paragraph.

---

## Author Comment (AC1) · 15 Dec 2020

**Answer to the referee 1**

First of all, we would like to thank the referee for her/his review of our paper and for giving us the opportunity to improve it.

The answer to the comments is organized as follows. First, we list some notations that will be adopted in the answers and the major changes done to the paper. Then, we detail our answers to the questions raised by the referee.

**Notations:**

- **Old version of the paper**: means the version submitted before.
- **New version of the paper**: means the version we submitted after the modifications based on the referee's comments.
- The '**_R1**' added in the legends of the figures: means referee 1.

**Changes:**

- We have removed Figure 3, 4 and 5 (of the old version) from the new version of the paper (the reason is detailed in the comment 3).
- We have modified Figure 7 (of the old version) to show data averaged by grid box as suggested by the referee.
- Table 1 was extended to include other lines.
- RefExp is replaced by RdiagExp in the new version of the paper and in the answers too.

**Answer to the questions of the referee:**

The question is copied in italic and the answer is written in normal font.

**I.   Underline{General Comments:}**

1. *A guideline is missing. What is the main objective of this paper? To diagnose an R-matrix or to improve ozone analyses using a diagnosed R-matrix?*

   **Answer**

   It is true that the way we have defined the main objective of this paper in the introduction (L31 P2 of the new version) might cause confusion between both estimating R-matrix or improving ozone analyses. As a matter of fact, the main objective is to improve the ozone analyses, by the mean of using more realistic observation error covariances. Estimating and discussing the R-matrix was not an end in itself, it is used to improve the assimilation of IASI radiances.
   In this new version, we have reduced the discussion of the diagnostic results (land/sea & day/night) to put much more emphasis on the main objective: ozone analyses.
   We modified the paper to include this comment while defining the objective (L31 P2 of the new version).

2. *The paper lacks a discussion about the bias correction that may be needed for ozone-sensitive channels. A comparison with the work of (Han and McNally, 2010) would have been relevant:*

   **Answer**

   We agree on the fact that the discussion of the bias correction was not well detailed in the paper. We give here more details, and we modified the paper to include this discussion (L8 P7).

   In NWP, the systematic errors in satellite observations are in general corrected before assimilating the observations or within the data assimilation process by VarBC scheme (Auligné et al., 2007). The key assumption is that the background state provided by the NWP system is unbiased. This assumption is not valid in atmospheric chemistry applications, where models might have significant biases, which is the case in our study (see figure 4 in Emili et al., 2019). In such case, VarBC requires some independent data (anchor) to prevent the drift of the analyses to unrealistic values that might be introduced by the model bias. In our case, we control tropospheric and stratospheric ozone. Identifying an anchor needs to be investigated carefully. Ozonesondes might be used as an anchor in the troposphere and low stratosphere, but the number of profiles provided is limited spatially and temporally. This might have an impact on the capacity of ozonesondes measurements to prevent the drift of the

analyses due to the model bias. Han et al. 2010, have used the channel 1585 (9.61μm) as an anchor in the assimilation of ozone for NWP. Dragani et al. 2013, have used the same uncorrected channel as anchor and they showed that its impact was not sufficient to stabilize the bias correction process for the long period. This aspect needs to be explored carefully in a separate study.

On the other side, a good understanding of sources of the measurements bias is a prerequisite to implement a bias correction scheme. VarBC in NWP applications, for instance, needs to define a linear model with some predictors (Auligné et al., 2007). Before adapting this approach in atmospheric chemistry framework, the possible sources of systematic errors in IASI ozone window need to be assessed.

In atmospheric chemistry, we were used to assimilate level 2 products of ozone (e.g. Massart et al., 2012; Emili et al., 2014; Peiro et al., 2018). Only recently, the direct assimilation of IASI radiances has been introduced in our chemistry transport model (Emili et al., 2019). Implementing a bias correction scheme requires careful diagnosis of the bias from observations monitoring. On the other hand, choosing an anchor demands also particular care and the choice depends on the full set of assimilated instruments. In this work, which is not based on a preexisting operational setup, we do not assimilate other ozone instruments than IASI. Thus, we had to assume that our observations are unbiased and we did not perform any bias correction. This assumption was adopted in many chemical analyses' studies before (e.g. Emili et al., 2019; Massart et al., 2012). Maintaining a similar framework allows a fairer comparison to these studies and might serve as a base for a future investigation of bias correction procedure for IASI.

We have modified the paper to include this discussion (L8 P7).

3.  *The comparisons between observation- errors according to surface types and between day and night are interesting:*

Here we want to remind the referee about a change that we have introduced to the new version of the paper following one of the referee2's comments (referee 2, 1st comment):

Since the separate treatment of land/sea covariance matrices did not yield significant results, we propose in this new version of the manuscript to keep only one paragraph discussing this aspect (L15 P10 to L10 P11 of the new version). As suggested by the referee 2, we cut the figures of day/night and sea/land (Figure 3, 4 and 5 of the old version). We gave more details about this choice in the answer to the comment 1 of the referee 2 answers.

We address the referee to check the referee2's answer for more details.

4.  *I wonder about the significance of an experiment of one month. It would have been beneficial to continue these experiments over 2 months, as well as over two distinct periods (summer and winter)?*

It is certainly true that the longer the period of the study, the more significant the results. However, our main objective was to verify if an update of the observations error can have an impact in the ozone analysis accuracy, and our reference analysis is the one-month experiment already discussed in Emili et al. (2019). We show in the paper that the impact is significant in terms of ozone concentration. We also show that scores are globally improved against three set of independent validation observations (ozonesondes, MLS and OMI) with very different coverage and accuracy during both summer and winter (northern and southern hemisphere). The statistical significance of these results for the month of July 2010 is hence ensured. Nevertheless, extending the period of the experiment is important to verify the robustness of the approach and it is one of our perspective for the future. Indeed, Emili et al., 2020, have used a correlated matrix (as in the paper) to assess the impact of IASI measurements on global ozone reanalysis for a duration of one year (personal communication, manuscript already submitted to Geoscientific Model Development).

**II. **Specific comments:**

1. *Specify in the title, that this work is carried out in a chemistry transport model.*

   **Answer:**

New title: 'Estimation of the error covariance matrix for IASI radiances and its impact on the assimilation of ozone in a chemistry transport model.'

2. *P1, L6: (...between 980 and 1100 cm-1) I suggest adding that this spectral range includes ozone-sensitive channels and atmospheric window channels.*

   **Answer:**

We used a subset of 280 channels…to estimate the observation error covariance matrix. This spectral range includes ozone-sensitive channels and atmospheric window channels. We computed hourly …

3. *P1, L11: (The computational cost...) This sentence is useless without explanation. I suggest you delete it or add a short comment.*

   **Answer:**

The computational cost was …in the assimilation system, by reducing the number of iterations needed for the minimizer to converge.

4. *P2, L30: (. . .impact on analysis accuracy.) Specify that this is the impact on the ozone analysis.*

   **Answer:**

   impact on the ozone analysis accuracy.

5. *P3, L2 There are more recent studies on the same subject that you can reference: (Weston et al. 2014, Bormann et al. 2016, Tabeart et al. 2020, Coopmann et al. 2020)*

   **Answer**

This line was modified to include some other references: (Weston et al. 2014, Bormann et al. 2016, Tabeart et al. 2020, Coopmann et al. 2020)

6. *P3, L29: (. . .the radiative transfer model RTTOV) Most recent reference to the work of (Saunders et al. 2018).*

   **Answer**

Saunders et al. 2018 was added in the references.

7. *P3, L31: (...Starting from an atmospheric...) Specify that RTTOV requires a vertical temperature and humidity profile.*

   **Answer**

Replace "Starting from an atmospheric vertical profile" by "Giving an atmospheric profile of temperature, water vapour and, optionally, trace gases, aerosols and hydrometeors, together with surface parameters and a viewing geometry", RTTOV simulates….

8. *P4, L6: What about other chemical variables (CO2, CH4, CO, N2O, SO2)? Do you use reference profiles? Which coefficient file do you take into account?*

   **Answer**

These chemical variables (CO2, CH4, CO, N2O) were set to the reference profiles of RTTOV. For the coefficient file, we used the coefficients for v9 predictors computed on 101 levels.
The SO2 was not available in RTTOV v11 (used in this study).
We have added this comment to the paper (L13 P4 of the new version)

9. *P3, L17: Indeed, the observation-error variances and observation-error covariances plays a fundamental role in the data assimilation process. In addition, background- errors are also very important in this process. For the purpose of consistency, it is required, at least, to show the background-error variances or background-error error standard deviation, as well as, the background-error correlations matrix.*

   **Answer:**

We have plotted the background-error standard deviation in % of the background profile (Figure 1_R1) and the zonal error correlation length scale Lx (Figure2_R1).

In the paper, we prefer not to show these figures since they are not very informative but we have added the background-error description in the table 1 of the new version of the paper.

[Figure]

| Figure 1_R1 : background-error standard deviation (square root of the diagonal of B) in % of the background profile. | Figure 2_R1: The zonal error correlation length scale (Lx). |

*10. P4, L19: (as a percentage of the observation values.) What does this percentage look like?*

**Answer:**

The referee is right, this sentence was not very clear. In the beginning of this paragraph, we wanted to list first all the possibilities offered by MOCAGE.

Since in our study we define our observation's error covariances in a file (as input), we omit this sentence and we keep only the case we are using (R-matrix read from a file).

This paragraph was omitted.

'In the data assimilation system of MOCAGE, the observation error covariance matrix can be read from the data file previously defined. In the case of diagonal matrix, the variances can be calculated as a percentage of the observation values.'

*11. P4, L22: Are there other variables included in the control vector?*

**Answer:**

No. The control vector contains only ozone and surface skin temperature. The word 'only' was added to this sentence on the paper.

*12. P5, Table 1: Can you provide more information about the ozone background?*

**Answer:**

This column was added to table 1.

| Ozone background | Hourly 3D forecasts of MOCAGE. |
|---|---|

*13. P5, L15: (. . .co-located land mask...) Wouldn't it be the "Land Sea Mask" instead?*

**Answer:**

Yes, this line was modified '…Data files also contain the co-located land sea mask and cloud fraction values…'

*14. P5, L16: In this case, from which satellite platform are IASI observations extracted? MetopA, B, C?*

**Answer**

Data are extracted from the MetopA platform. MetopB and C were not available for the period of the study.

*15. P6, L20: Another reference to (Emili et al. 2019) It would be very useful to summarize the configuration of the experiments in a table.*

**Answer**

In this new version of the paper, we extend Table 1 to include other elements of the experiment's configuration.

Below, we present lines added to the Table 1 of the paper.

| Period of the study | July 2010 |
|---|---|
| Background error | Vertically variable and computed as % of the background profile (using a value of 2% above 50 hPa and 10 % below). |
| Background error zonal correlation | Exponential with a length scale set to 200 Km and reduced towards the pole to account for the |

| | increasing zonal resolution of the regular latitude-longitude grid. |
|---|---|
| Background meridional error correlation | Exponential with a length scale set to 200 Km. |
| Background error vertical correlation | Exponential with a length scale set to 1 grid point (vertical level). |

16. *P6, L27: Can you compare these ozone background-error standard deviation with other values used in recent research?*

**Answer:**

The ozone background-error standard deviation was taken as percentages of the background $O_3$ profile. This strategy was adopted previously by many studies (e.g. Emili et al., 2014, Peiro et al., 2018, and Emili et al., 2019). Emili et al., 2014 and Peiro et al., 2018 have used a percentage of 15% in the troposphere and 5% in the stratosphere.

In this study, we have adopted a detailed chemical scheme (chemical scheme combining both Regional Atmospheric Chemistry Mechanism for the troposphere (Stockwell et al., 1997) and REPROBUS (Lefevre et al., 1994) for the stratosphere). This scheme was shown to reduce the model bias compared to scheme used in Emili et al., 2014 and Peiro et al., 2018 (see Figure 4 in Emili et., 2019). Hence, we chose the same background errors as in Emili et., 2019: 2% of the $O_3$ profile above 50hPa and 10% below. An important reason to keep the background errors similar to the setup of Emili et al. (2019) is also that we wanted to examine here exclusively the impact of **R**, as already reminded in the introduction and in the conclusion.

The paper was modified to add this discussion (L16 to L22 P6).

This part of the paper (P6 L27 to L29 of the old version):

"The background standard deviation was, thus, taken equal to 2% above 50 hPa and 10 % below to mimic the validation's behavior. Similar choices were employed in (Massart et al., 2012; Peiro et al., 2018).

Was replaced by (L16 P29 of the new version) by:

"This strategy was adopted previously by many studies (e.g. Emili et al., 2014, Peiro et al., 2018, and Emili et al., 2019). Emili et al., 2014 and Peiro et al., 2018 have used a percentage of 15% in the troposphere and 5% in the stratosphere.

In this study, we have adopted a detailed chemical scheme. This scheme was shown to reduce the model bias compared to scheme used in Emili et al., 2014 and Peiro et al., 2018 (see Figure 4 in Emili et., 2019). Hence, we chose the same background errors as in Emili et., 2019: 2% of the $O_3$ profile above

50hPa and 10% below. An important reason to keep the background errors similar to the setup of Emili et al. (2019) is also that we wanted to examine here exclusively the impact of R, as already reminded in the introduction and in the conclusion."

17. P7, L11: On what criteria were these channels identified as sensitive to water vapor?

The channels sensitive to water vapor in the ozone band have been identified by previous studies on IASI trace gases retrieval using RT simulations (Barret et al 2011, see also Fig. 1 of https://acp.copernicus.org/articles/11/857/2011/acp-11-857-2011.pdf ). We used here the same channel selection of previous O3 studies.

*18. P8, L18 to L28: This paragraph is complicated to follow and it is a pity because it is important for the next step. I suggest you summarize the different configurations in a table.*

We have redrafted this paragraph (L15 to L24 P9 of the new version of paper) to improve the clarity of the discussion of different estimations and the one used in the paper. This paragraph was replaced by:

"Using outputs (analyses and forecasts) derived from 3D-Var experiment that uses a diagonal R-matrix (called hereafter 1st 3D-Var experiment) in the estimation process might have an impact on the diagnosed R-matrix. The matrix derived using these outputs is called hereafter 1st estimation. We performed another 3D-Var experiment (2nd 3D-Var experiment) using the 1st estimation. The outputs (analyses and forecasts) of this experiment (2nd 3D-Var experiment) were used to estimate another R-matrix called 2nd estimation. The standard deviation of the 2nd estimation is larger than that of the 1st estimation (not shown). The same goes for correlations (not shown). It should be noted that the 2nd estimation was positive definite, unlike the 1st estimation where some unrealistic features were encountered. We have followed the same process to reestimate two other matrices (3rd and 4th estimation). The differences of the estimations in terms of standard deviation and correlations became smaller as we reestimate the matrices, suggesting a sort of convergence of the estimation.

We have adopted the 2nd estimation for the results shown in this work. The reason for this choice will be discussed later (section 5.2)."

We show below a figure summarizing this discussion. This figure was not added to the paper.

[Figure]

Figure 3_R1 : different estimations discussed in the paper and the one used for the results.

19. *P10, Figure 2: Correlation matrices can vary between -1.0 and 1.0. I expected to see negative correlations between some channels in the atmospheric window and some ozone-sensitive channels. Why not represent the matrix between -1.0 and 1.0, centered on white at zero?*

Since the ozone-sensitive and SST-sensitive channels present high interchannel correlations in this spectral window, we set the limits of the correlations between 0.3 and 1 to improve the information content of the figures. Also, no negative

values were encountered. We present below Figure 4_R1 (the same as Figure 2 of the old version of the paper) with -1.0 and 1.0 as limits:

[Figure]

Figure 4_R2: Correlation matrix estimated over the globe (sea and land).

20. *P12, Figure 5: Same remark as above about the color scale.*

No negative correlations have been encountered in Figure 5.a, 5.b, and 5.c (of the old version of the paper). In fact, since the ozone-sensitive and SST-sensitive channels present high interchannel correlations, we set the limits of the correlations between 0.3 and 1. For Figure 5.e and Figure5.d (the differences in the old version of the paper) we took the absolute value of the differences divided by the global estimation.

We show below Figure 5_R1 (5b_R1, 5c_R1) the same Figure 5 (b and c) of the old version of the paper with -1.0 and 1.0 as limits (Figure 5 a of the old paper is the same in the previous comment 19). Please note that the Figure 5 (of old version) was removed from this new version of the paper.

[Figure]

| Figure 5b_R1: Correlation matrix estimated over the sea. | Figure 5c_R1 : Correlation matrix estimated over the land. |

21. *P12, L8: The naming of the experiments is not appropriate because one could confuse Control and Reference. I would suggest RdiagExp instead of RefExp.*

**Answer:**

RefExp is changed to RdiagExp

22. *P14, L4: It would be useful to explain the physical link between skin temperature and ozone in the assimilation of infrared observations. Is there any consideration of inter- variable background-error correlations between O3 and Tskin?*

**Answer:**

Yes, indeed. The skin temperature is physically linked to the ozone measured. In fact, the skin temperature interacts with the ambient atmosphere. An increase of SST can for example create a convective movement impacting the transport of the ozone. However, the skin temperature is given only at the observation location in this study and it is specified with values interpolated from NWP forecasts (IFS), whereas ozone is a 3D field issued from the chemistry transport model. Hence, the estimation and potential account of error correlations between the two variables seems challenging in our system. We think that Earth System models where both skin temperature and ozone are modeled (and assimilated) might represent a preferable framework for analyzing this particular aspect.

In this work, we did not consider the background-error correlation that might exist between O3 and SST.
We have modified the paper to include this paragraph (L6 P12 of the new version of the paper).

23. *P14, Figure 7: There is also increase in difference on land using RfullExp, mainly in Africa and South America. This can be related to the differences in observation-errors depending on the surface. . . In addition, there are too many pixels on the map. It would be interesting to average by box in order to better exploit the information provided by this Figure.*

  23.1. *There is also increase in difference on land using RfullExp, mainly in Africa and South America:*

   **Answer:**

   Indeed, the increase in difference over the land seems related to the dependence of observation-errors on the surface. In fact, the number of observations over the sea represents almost 70% of the total observations we have used in this study. Consequently, our SST analysis stays closer to background values (IFS forecasts) over the sea than over the land.
   This comment was added to the paper (L4 P13).

  23.2. *. It would be interesting to average by box in order to better exploit the information provided by this Figure.*

   **Answer:**

Figure 7 (of the old version) is replaced by other figures below where we have averaged the observations by grid box.

[Figure]

Fig7a_R1. Difference (in °C) between the IFS SST forecast and the analysis of the SST given by RdiagExp (averaged by grid box).

[Figure]

Fig7b_R1. Difference (in °C) between the IFS SST forecast and the analysis of the SST given by RfullExp (average by grid box).

24. *IASI channels between 1000 and 1070 cm-1 are mainly sensitive to ozone above 100 hPa, which poses the challenge of using other observations for a complete analysis of ozone over the entire atmospheric column...*

Yes, indeed. It would have been advantageous to assimilate other instruments (MLS for example in the stratosphere and ozonesondes for the free troposphere) for a complete analysis of ozone. However, we wanted to evaluate, through this study, the impact of accounting for interchannel error correlations of IASI in the assimilation system. Assimilating other accurate instruments might alleviate (or hide) the impact of interchannel observation-error correlations of IASI on the analysis, as it was shown in Emili et., al (2019).

25. *the high sensitivity of the ozone channels raises the problem of the amount of information remaining after a cloud detection...*

Yes, it would be challenging to take into account the pixels affected by clouds (and including the corresponding cloud properties in the radiative transfer) during the assimilation of IASI channels for ozone. This might be an area of research for future work.

The idea for this study was to keep the same configuration of the assimilation system adopted in the study of Emili et al., 2019, to be able to evaluate only the impact of an updated observation error covariance matrix in clear-sky conditions.

**III.    **Technical comments:**

1. *P1, L4: (Modèle de Chimie Atmosphérique à Grande Echelle)*

   **Corrected**

2. *Throughout the paper: I suggest (Chemistry Transport Model) instead of (Chemical transport model)*

   *P1, L5: (. . .already adopted in numerical weather prediction centers) This is not the case for all centers, (. . .already adopted in some numerical weather prediction centers)*

   **Corrected**

3. *Throughout the paper: Beware of the systematic use of (Furthermore). Vary the ad- verbs.*

   **Corrected**

4. *P2, L8: (. . .to construct a realistic picture of the...) The term (picture) is not appropriate, I suggest changing the word.*

   **Corrected**

5. *P2, L22: (...uncorrelated, some Numerical Weather Prediction...) P2, L31: (. . .evaluate their impact on the ozone analysis accuracy) P3, L19: (. . .MOCAGE is fed with forced by meteorological...)*

   **Corrected**

6. *P5, L5: (. . .the polar-orbiting satellite Metop-A, B and C launched...)*

   **Corrected**

7. *P6, L24: (The ozone forecast-error standard deviation...)*

   **Corrected**

8. *P6, L27: (The ozone background-error standard deviation...)*

   **Corrected**

9. *P7, L1: (The ozone background-error covariance matrix. . .)*

   **Corrected**

10. *P9, L12: (. . .we present the diagnosed correlation matrix...)*

    **Corrected**

11. *Throughout the paper: Be careful to capitalize the words (Figure)*

    **Corrected**

12. *Throughout the paper: Write rather with dashes (observation-errors, background- errors , ozone-sensitive,...)*

    **Corrected**

13. *P15, L22: (. . .encountered in these regions in the stratosphere...)*

    **Corrected**

Auligné T, McNally AP, Dee DP. 2007. Adaptive bias correction for satellite data in a numerical weather prediction system. *Q. J. R. Meteorol. Soc.* **133**: 631–642

Bormann, N., Bonavita, M., Dragani, R., Eresmaa, R., Matricardi, M., and Mcnally, T.: Observations Through an Updated Observation Error Covariance Matrix, 2015.

Coopmann, O., Guidard, V., Fourrié, N., Josse, B., and Marécal, V.: Update of Infrared Atmospheric Sounding Interferometer (IASI) channel selection with correlated observation errors for numerical weather prediction (NWP), Atmospheric Measurement Techniques, 13, 2659– 2680, https://doi.org/10.5194/amt-13-2659-2020, 2020.

Dragani R, McNally AP. 2013. Operational assimilation of ozone-sensitive infrared radiances at ECMWF. *Q. J. R. Meteorol. Soc.* **139**: 2068–2080. DOI:10.1002/qj.2106

Emili, E., Barret, B., Le Flochmoën, E., and Cariolle, D.: Comparison between the assimilation of IASI Level 2 retrievals and Level 1 radiances for ozone reanalyses, Atmospheric Measurement Techniques Discussions, pp. 1–28, https://doi.org/10.5194/amt-2018-426, 2019.

Han W, McNally AP. 2010. The 4D-Var assimilation of ozone-sensitive infrared radiances measured by IASI. *Q. J. R. Meteorol. Soc.* **136**: 2025 – 2037.

Lefevre et al., 1994] Lefevre, F., Brasseur, G. P., Folkins, L., Smith, A. K., and Simon, P. (1994). Chemistry of the 1991-1992 stratospheric winter : Three- dimensional model simulations. *Journal of Geophysical Research*, 99(D4):8183– 8195.

Massart, S., Piacentini, A., and Pannekoucke, O.: Importance of using ensemble estimated background error covariances for the quality of atmospheric ozone analyses, Quarterly Journal of the Royal Meteorological Society, 138, 889–905, https://doi.org/10.1002/qj.971, 2012.

Peiro, H., Emili, E., Cariolle, D., Barret, B., and Le Flochmoën, E.: Multi-year assimilation of IASI and MLS ozone retrievals: Variability of tropospheric ozone over the tropics in response to ENSO, Atmospheric Chemistry and Physics, 18, 6939–6958,

Saunders, R., Hocking, J., Turner, E., Rayer, P., Rundle, D., Brunel, P., Vidot, J., Roquet, P., Matricardi, M., Geer, A., Bormann, N., and Lupu, C.: An update on the RTTOV fast radiative transfer model (currently at version 12), Geoscientific Model Development, 11, 2717–2737, https://doi.org/10.5194/gmd-11-2717-2018, 2018.

Stewart, L. M., Dance, S. L., Nichols, N. K., Eyre, J. R., and Cameron, J.: Estimating interchannel observation-error correlations for IASI radiance data in the Met Office system, Quarterly Journal of the Royal Meteorological Society, 140, 1236–1244, https://doi.org/10.1002/qj.2211, 2014

Stockwell et al., 1997] Stockwell, W. R., Kirchner, F., Kuhn, M., and Seefeld, S. (1997). A new mechanism for regional atmospheric chemistry modeling. *Journal of Geophysical Research*, 102:25847–25879.

Tabeart, J. M., Dance, S. L., Lawless, A. S., Migliorini, S., Nichols, N. K., Smith, F., and Waller, J. A.: The impact of using reconditioned correlated observation-error covariance matrices in the Met Office 1D-Var system, Quarterly Journal of the Royal Meteorological Society, pp. 1–22, https://doi.org/10.1002/qj.3741, 2020a.

---

## Author Comment (AC2) · 15 Dec 2020

**Answer to the referee 2**

First of all, we would like to thank the referee for her/his review of our paper and for giving us the opportunity to improve our paper.

   The answer to the comments is organized as follows. First, we list some notations that will be adopted in the answers and the main changes done to the paper. Then, we detail our answers to the questions raised by the referee.

**Notations:**

- **Old version of the paper**: means the version submitted before.
- **New version of the paper**: means the version we submitted after the modifications based on the referee's comments.
- The '**_R2'** added in the legend of the figures: means referee 2.

**Changes:**

- We have removed Figure 3, 4 and 5 (of the old version) from the new version of the paper (the reason is detailed in the comment 1).
- We have modified Figure 7 (of the old version) to show data averaged by grid box as suggested by the referee 1 (see comment 23 of general comments of referee 1).
- Table 1 was extended to include other lines.
- RefExp is replaced by RdiagExp in the new version of the paper and in the answers too.

**Answer to the questions of the referee:**

The question is copied in italic and the answer is written in normal font.

**I.     Underlined: General Comments:**

1. *I would welcome a discussion on why the separate treatment of the land/sea covariance matrices did not yield significant results. Otherwise I think the land/sea and day/night results could be cut:*

   **Answer**

   Since the separate treatment of land/sea covariance matrices did not yield significant results, we propose in this new version of the manuscript to keep only one paragraph discussing this aspect (L15 P10 to L10 P11 of the new version). As suggested in the comment of the referee, we cut the figures of day/night and sea/land discussion (Figure 3, 4 and 5 of the old version of the paper). We give here more details for this choice.

   As we pointed out in the first version of the submitted paper, the separation of the type of the surface of observations during the assimilation did not show a significant difference with the case of considering a global estimation.

   Figure 1_R2 shows the relative difference of the RMSE with respect to radiosoundings (see the formulation used to compute them in comment 21.2 of specific comments) for an experiment using the R estimated globally (green line), R estimated with separation of the type surface (red line, where each pixel is attached to a matrix estimated according to the type of the its surface (land/sea)), and R diagonal (blue line). The same validation was adopted in figure 9 of the paper (old version). The validation against ozonesondes shows slight differences with small improvements around 150 hPa in the tropics and in the southernvmidlatitudes (30S-60S) free troposphere.

   Figure 2_R2 shows the same results reported in the figure 8 of the manuscript (old version): Difference of the ozone total column (DU) provided by OMI and that of the assimilation experiment using an estimated R-matrix globally, averaged over the month of the study.

   Figure 3_R2 shows the difference of the ozone total column (DU) provided by OMI and that of the assimilation experiment using an estimated R-matrix according to the type of the surface (land/sea) averaged over the month of the study.

   A comparison between Figure 2_R2 and Figure 3_R2 shows that the separation of the type of the observation's surface type did not yield significant differences in terms of total column.

This behavior might be explained by the number of observations over the sea and over the land. In fact, the observations over the sea represent more than 70% of the total of observations. As we can notice in Figure 4_R2, the differences, in terms of standard deviation, of the global estimation and that using pixels over the sea is very small in comparison with that using pixels over the land. The differences are also small in terms of correlations in the case of the sea surface in comparison with the land surface (Figure 5_R2). Hence, we consider that the predominance of observations over sea averages out the potential differences caused by a separate land/sea specification of R. This papragraph was added to the paper (L27 P16 of the naw version of the paper)

As it was suggested by the referee's comment, we removed the figures of day/night and sea/land comparison (Figure 3, 4 and 5 of the old version of the paper) and we keep only one paragraph (L15 P10 to L10 P11 of the new version) that resumes this discussion.

[Figure]

Figure 1_R2: Normalized difference of the RMSE with respect to radiosoundings for the RFullExp (green), RdiagExp (blue) and the RfullExp_LS (the experiment using separated matrix according to the type of the surface) in red. The difference of the RMSE was computed by subtracting the RMSE of the controlExp from the RMSE of the analysis of each experiment, divided by the average profile of radiosoundings (see the formulation in comment 21.2 of specific comments).

[Figure]

[Figure]

Figure 2_R2: Difference of the ozone total column (DU) provided by OMI and that of the assimilation experiment using a matrix estimated over the glob averaged over the month of the study.

Figure 3_R2: Difference of the ozone total column (DU) provided by OMI and that of the assimilation experiment using a matrix according to the type of the surface (land/sea) averaged over the month of the study.

[Figure]

[Figure]

[Figure]

Figure 4_R2: Standard deviation estimated using Desroziers diagnostics according to the type of the surface (sea, land and global).

Figure 5_R2: Difference (in %) between global and sea correlation matrix (divided by the global matrix).

Figure 6_R2: Difference (in %) between global and land correlation matrix (divided by the global matrix).

2. *Condition number discussion*

   *2.1. It seems that reconditioning was only used to correct negative values which could result in a nearly singular matrix?*

   Yes, in fact the conditioning method was based on the correction of the negative values. The objective was to obtain a symmetric positive definite matrix. The resulting matrix shows, indeed, very strong interchannel error correlations and remains relatively ill-conditioned. Nonetheless, to ensure that the inversion of the matrix was performed correctly, we computed the product of the R-matrix and its inverse and we checked that it is equal to the identity (with a precision of $10^{-4}$), before any further use within the assimilation.

   *2.2. What is the minimization algorithm? Does it include a preconditioner that depends on R?*

   The minimization algorithm used in this work is LBFGS (Liu et al., 1989).

   No, it does not include a preconditioner that depends on R. But, the system is preconditioned with the square root of the B-matrix.

   The paper was modified to include this comment.

   2.3. *What are the final condition number?*

   The majority of eigenvalues are very small comparing to the maximum value. That makes it a bit difficult to get a well-conditioned matrix without changing dramatically the matrix. We have verified that our matrix was well inverted before any further use within the assimilation.
   The final condition number is: $8. \ 10^6$.

3. *Was bias correction used and if so, what method? If not, were significant biases observed in the IASI observations?*

   3.1. *Was bias correction used and if so, what method?*

   **Answer**

   No bias correction was used in this study. A detailed discussion is given in the response to R1 and reported here for completeness.

   In NWP, the systematic errors in satellite observations are in general corrected before assimilating the observations or within the data assimilation process by VarBC scheme (Auligné et al., 2007). The key assumption is that the background state provided by the NWP system is unbiased. This assumption is not valid in atmospheric chemistry applications, where models might have significant biases, which is the case in our study (see figure 4 in Emili et al., 2019). In such case, VarBC requires some independent data (anchor) to prevent the drift of the analyses to unrealistic values that might be introduced by the model bias. In our case, we control tropospheric and stratospheric ozone. Identifying an anchor needs to be investigated carefully.

Ozonesondes might be used as an anchor in the troposphere and low stratosphere, but the number of profiles provided is limited spatially and temporally. This might have an impact on the capacity of ozonesondes measurements to prevent the drift of the analyses due to the model bias. Han et al. 2010, have used the channel 1585 (9.61μm) as an anchor in the assimilation of ozone for NWP. Dragani et al. 2013, have used the same uncorrected channel as anchor and they showed that its impact was not sufficient to stabilize the bias correction process for the long period. This aspect needs to be explored carefully in a separate study.

On the other side, a good understanding of sources of the measurements bias is a prerequisite to implement a bias correction scheme. VarBC in NWP applications, for instance, needs to define a linear model with some predictors (Auligné et al., 2007). Before adapting this approach in atmospheric chemistry framework, the possible sources of systematic errors in IASI ozone window need to be assessed.

In atmospheric chemistry, we were used to assimilate level 2 products of ozone (e.g. Massart et al., 2012; Emili et al., 2014; Peiro et al., 2018). Only recently, the direct assimilation of IASI radiances has been introduced in our chemistry transport model (Emili et al., 2019). Implementing a bias correction scheme requires careful diagnosis of the bias from observations monitoring. On the other hand, choosing an anchor demands also particular care and the choice depends on the full set of assimilated instruments. In this work, which is not based on a preexisting operational setup, we do not assimilate other ozone instruments than IASI. Thus, we had to assume that our observations are unbiased and we did not perform any bias correction. This assumption was adopted in many chemical analyses' studies before (e.g. Emili et al., 2019; Massart et al., 2012). Maintaining a similar framework allows a fairer comparison to these studies and might serve as a base for a future investigation of bias correction procedure for IASI.

We have modified the paper to include this discussion (L8 P7).

3.2. *were significant biases observed in the IASI observations?*

It is true that to properly answer this question, we need to compare our observations to a set of independent data. These data might be derived from other instruments or from models assumed to be unbiased. In our case, the absence of an independent instrument's data that are dense enough (spatially and temporally), on one hand, and the model that is biased on the other hand (see Figure 4 in Emili et al., 2019) make this approach (comparison with independent data) difficult to perform. To give a broad view of the bias in our system, we suggest here to show observation-background (O-B) statistics for six separate channels picked arbitrarily from the used band.

Figure 7_R2 shows the O-B statistics (averaged daily) as a percentage of the daily average of observations of each channel. We note that O-B varies between 0.6% and 1.7% over the observations for the six channels showed here. The same behavior of O-B statistics as percenatge of the background of each channel can be observed in Figure 8_R2. Thus, we can conclude that O-B is not too large compared to the background and observations values.

Nevertheless, the O-B statistics might not reflect a real bias of IASI observations since our model can biased. To address carefully this question and detect the bias of the IASI observations, an independent study is required. As we pointed above (comment 3.1), to maintain the same framework of the previous works (Emili et al., 2019)  and aiming to evaluate

only the contribution of the observation-error covariances, we have assumed that our observations are unbiased. Yet, the bias correction procedure of IASI observations in our case (atmospheric chemistry) should be investigated in future work.

[Figure]

Figure 7_R2: The percentage of O-B statistics (averaged daily) over the daily average of observations of each channel (1414, 1505, 1553, 1585, 1633, 1671).

[Figure]

Figure 8_R2: The percentage of O-B statistics (averaged daily) over the daily average of the correspondent background of each channel (1414, 1505, 1553, 1585, 1633, 1671).

4. *The assimilation experiments were run for a 1-month period. Is this long enough to quantify the significance?*

It is certainly true that the longer the period of the study, the more significant the results. However, our main objective was to verify if an update of the observations error can have an impact in the ozone analysis accuracy, and our reference analysis is the one-month experiment already discussed in Emili et al. (2019). We show in the paper that the impact is significant in terms of ozone concentration. We also show that scores are globally improved against three set of independent validation observations (ozonesondes, MLS and OMI) with very different coverage and accuracy during both summer and winter (northern and southern hemisphere). The statistical significance of these results for the month of July 2010 is hence ensured. Nevertheless, extending the period of the experiment is important to verify the robustness of the approach and it is one of our perspective for the future. Indeed, Emili et al., 2020, have used a correlated matrix (as in the paper) to assess the impact of IASI measurements on global ozone reanalysis for a duration of one year (personal communication, manuscript already submitted to Geoscientific Model Development).

5. *Are SBUV available for assimilation or validation?*

Indeed, the SBUV and MLS might be assimilated for more accurate ozone analysis, at least in the stratosphere. However, we wanted to evaluate, through this study, the impact of accounting for interchannel error correlations of IASI in the assimilation system. Considering other accurate instruments might alleviate (or hide) the impact of IASI error covariance matrix on the analysis (as shown by Emili et al. 2019). For the validation we chose to validate with MLS rather than SBUV since it provides a continuous (during day and night) monitoring of the ozone as the infrared measurements with better vertical resolution.

**II. **Specific comments:**

1. *P2 L4 and P5 L5, IASI is on Metop-A, B and C. But only Metop-A was available during the period of this study.*

This comment was included, we have specified that data from Metop-A were used. MetopB and C were not available in 2010 (the period of the study).

2. *P2 L16 Other references exist that discuss sources of IR error and error correlations. Representivity errors can also contribute to inter-channel error correlations.*

This part of the paper was modified to include some other references:

"Bormann et al. (2009) has listed …. of quality control in the data assimilation system" is replaced by:

"The interchannel error correlations might originate from observation operator errors. They can also arise from the instrument calibration and some practices of quality control (Bormann et al. (2009), Waller et al. (2016), Geer at al., (2019)). The representation errors (e.g. Janjić et al., 2018) may also introduce correlations."

3. *P2 L31 Is the main objective to study the impact on ozone analysis accuracy? Also, I think it is worth mentioning here that this is within the framework of a CTM.*

Yes, the main objective of our work is to improve the ozone analysis. We, therefore, modified the introduction (L31 P2 of the new version) to include this comment:

The sentence: 'The main objective of this study is, thus, to estimate the observation error covariances for IASI ozone-sensitive channels and to evaluate their impact on the analysis accuracy' is replaced by:

'The main objective of this study is, thus, to improve the ozone analysis accuracy within a chemistry transport model, by the main of using more realistic observation error covariances for IASI ozone-sensitive channels.

'their impact on the analysis accuracy' is replaced by 'their impact on the ozone assimilation within a chemistry transport model'

4. *P3 L1 There are numerous other studies that could be cited here.*

This line was modified to include some other references: (Weston et al. 2014, Bormann et al. 2016, Tabeart et al. 2020, Coopmann et al. 2020)

5. *P4 L22 Does the control vector include any other variables?*

No, it includes only SST and ozone. The paper was modified to precise that the control vector includes only SST and ozone.

This sentence: "The control vector includes the Skin Surface Temperature (SST) and the ozone." was replaced by:

"The control vector includes only the Skin Surface Temperature (SST) and the ozone."

6. *P5 L16. A brief discussion of channel selection is warranted. The abstract mentions that 280 channels are used, but this is worth restating here.*

We have added the band used:

"For this study, L1c data have been downloaded…" is replaced by

"For this study, a subset of 280 channels covering the spectral range between 980 and 1100 $cm^{-1}$ was used. The channel selection is inherited from IASI Level 2 O3 retrievals (Dufour et al. 2011, Emili et al. 2019). L1c data have been downloaded…."

7. *P6 L21 What observations are being assimilated? Assimilating observations from OMI, SBUV or ozonesondes could help anchor the bias correction of IASI ozone channels.*

We have assimilated only IASI data. We have used OMI and ozonesondes to validate our results. Yes, a combination of ozonesondes and SBUV or OMI might serve as an anchor while processing the bias correction. However, we have assumed that our observations are unbiased as in many previous studies, and we have discussed this choice in bias correction comment (comment 3).

8. *P7 L18 Change "missing" to "absence." Is there a better justification for this assumption?*

This paragraph was entirely modified to include the bias correction discussion introduced in comment 3 (General comments).

9. *P8 L1 There are many other references that should be cited in addition to Stewart et al, 2009. In addition, there have been a few theoretical studies on the Desroziers method that could be cited here.*

This line was modified to include some other references: (Bormann et al. 2016, Waller et al., 2016, Tabeart et al. 2020, Coopmann et al. 2020)

10. *P8 L8 How many days of data were used in the computation? How many days of data were used for the re-estimations?*

We have used the same month (July) of the study for the re-estimation. In fact, the objective was to avoid to use an analysis that came from a diagonal R-matrix.

11. *P8 L9 The term "positive definite" is typically not used when discussing asymmetric matrices. A symmetric matrix is positive definite if and only if all of its eigenvalues are positive. I suggest that you instead discuss the eigenvalues after the matrix is made symmetric.*

Actually, we have made the estimated matrix symmetric by adding to it its transpose divided by 2: $(0,5* (R+R^T))$. Then we started to discuss the eigenvalues.

12. *P8 L14 Citing Van Loan seems out of place here. Is there a specific page number? This again is an incorrect citation, Gene Golub was another author of this textbook. I believe this method was used in Weston et al 2014 and in Tabeart et al, 2020b, so these might be better references.*

Yes, the referee is wright. We have inverted, by mistake, the citations here. Also, Van Loan is wrongly cited.

We have modified this part to include this comment.

This part:

Then we impose the negative eigenvalues to be equal to the smallest positive eigenvalue (Charles F. 15 Van Loan, 1996). Another method was tested here to recondition the estimated matrix, the one called *ridge regression* (Weston et al., 2013; Tabeart et al., 2020b) which consists of increasing all eigenvalues of R by the same amount. We favored the first method since the standard deviation and the correlation values remain closer to the initially estimated quantities.

Was replaced by (L11 P9 of the new version):

Then we impose the negative eigenvalues to be equal to the smallest positive eigenvalue as in Weston et al., 2013 and Tabeart et al., 2020b. Another method which consists of increasing all eigenvalues of R by the same amount was tested. We favored the first method since the standard deviation and the correlation values remain closer to the initially estimated quantities.

13. *P8 L22 Positive definite- was the re-estimated matrix symmetric?*

No, we have made it symmetric by adding to it its transpose divided by 2: $(0,5* (R+R^T))$.

14. *Figure 1- Please state in the caption what the "previous studies" are.*

Emili et al. 2019 was added to the Figure.

15. *Figure 2, 5- I do not trust the tick labels on the x and y axes. In Figure 1, there seems to be a gap in channels between 1010 and 1020 cm-1. If the ticks on Figure 2 are linear then the plotting program might be interpolating the covariances between 1010-1020 cm-1, or the tick labels could be wrong. I know that Matlab for example does not label ticks correctly by default when making matrix plots like these.*

Indeed, the tick labels were wrong. We modified this figure and we show the result below.

[Figure]

Figure 9_R2: R-matrix estimation over the globe (sea and land).

16. *in the caption, "statistics of Desroziers" would read better as "Desroziers method" or "Desroziers diagnostic"*

  "statistics of Desroziers" is replaced by "Desroziers method"

17. *Figure 5- Anti-correlations likely exist over land. I suggest changing the colorbar scale to include negative values.*

In fact, since the ozone-sensitive and SST-sensitive channels present high interchannel correlations in this spectral window, we set the limits of the correlations between 0.3 and 1 to improve the information content of the figures. Also, no negative values were encountered in Figure 5.a, 5.b, and 5.c (of the old version of the paper). For Figure 5.e and Figure5.d (the differences in the old version of the paper) we took the absolute value of the differences divided by the global estimation.

We present below Figure 10_R2 (10b_R2, 10c_R2) the same Figure 5 (b and c) of the old version of the paper with -1.0 and 1.0 as limits. The same behavior was encountered in the Figure 5a (of the old version). Please note that the Figure 5 (of old version) was removed from this new version of the paper.

[Figure]

| Figure 10b_R2: Estimation over the sea. | Figure 10c_R2: Estimation over the land. |

18. *P13 L7 Suggest to specify in this sentence that this is an ozone analysis.*

… between the zonal average of the ozone analysis

19. *P13 L14 Why is the reduction important?*

In fact, by 'important' we wanted to say 'large'. We have replaced important by 'large'.

20. *P15 L10-14. I do not understand this discussion. What is the "estimation?" Why did the minimization fail to converge when using the "first estimation" but not the "second estimation?"*

Indeed, this discussion was not written in a way which allows a clear understanding. The first estimation did not fail to converge but needs more iterations than other estimations to converge.

We give, below, more details about what we meant by this discussion. We modified the paper to include this comment (L4 P14 to L13 P14).

We wanted to discern the imapct of the variance of that of the correlations on the convergence speed. To this end, we have performed three assimilation experiments using different R-matrices:

1st experiment: the employed R is estimated from the analysis computed using a diagonal R. The minimizer takes generally more than 100 iterations to converge.

2nd experiment: We use the analysis given by the 1st experiment to estimate another R-matrix (called second estimation in the old version of the paper). We have used this estimation to run another assimilation cycle. We have noticed that the minimizer needs about 60 iterations to converge.

3rd experiment: We have modified the R-matrix of the first experiment: we kept its correlations and replace its standard deviation with that of R used in the second experiment. We have noticed that the minimizer needs less than 100 iterations to converge (about 70 iterations).

Actually, using the first estimation of R, the minimization needs more than 100 iterations to converge, whereas about 60 iterations are needed with the second estimation of R. The results of the 3$^{rd}$ experiment seem to suggest that updating the variance has a larger impact on the convergence.

21. *Figures 9 and 10, in the captions it would be helpful to state that negative values indicate an improvement in fit relative to ControlExp. Is it possible to remove the empty space in Figure 10? What is meant by "divided by the average profile of radiosoundings?"*

   21.1. *Figures 9 and 10, in the captions it would be helpful to state that negative values indicate an improvement in fit relative to ControlExp. Is it possible to remove the empty space in Figure 10?*

   The figure was modified and included in the paper.

   21.2.  *What is meant by "divided by the average profile of radiosoundings?"*

   We have replaced 'relative' by 'normalized' in the new version of the paper. We remind here how we computed RMSE presented in the figure 6 and 7 (of the new version of the paper).

   - Figure 6:  | **(RMSE (control) − RMSE (exp))/radiosoundings** |

   - Figure 7:  | **(RMSE (control) − RMSE (exp))/MLS** |

   Where: - 'exp': stand for RdiagExp (blue) and for RfullExp (green).

   - 'radiosoundings': the average profile of the ozonesondes.

   - 'MLS': the average of the MLS profiles.

   - 'control': control experiment.

[Figure]

Fig 6_R2: Normalized difference of the RMSE with respect to the MLS for the R estimated (green) and R diagonal (blue). The normalized difference of the RMSE was computed by subtracting the RMSE of the analysis from the RMSE of the control for each experiment, divided by the average profile of the MLS. Negative values mean that the assimilation improved (decreased) the RMSE of the control simulation, and positive values indicate degradation (increase) of the RMSE

22. *P20 L16 Specify that ozone is added/reduced in the analysis.*

The word 'analysis' was added to specify that is the ozone analysis.

23. *P20 L17 "The total column also. . ." This sentence can be omitted. Isn't the total column result the validation against OMI mentioned in the next sentence?*

Yes, Indeed. The sentence is omitted.

**III.  Technical comments:**

1. *The work from Desroziers is not cited correctly. It is from 2005, not 2006. Weston et al. is cited incorrectly as well; this paper is from 2014.*

   **Corrected**

2. *Throughout the article, ozone is written as O3, when it should be O3 with the 3 in the subscript.*

   **Corrected**

3. *P1 L22 Change "Remote sounding from satellites is" to "Remote soundings from satellites are"*

   **Corrected**

4. *P2 L1 Change "monitoring atmospheric gases, a large" to "monitoring of atmospheric gases. A large"*

   **Corrected**

5. *P2 L10 Should "Recent studies" be "A recent study" instead?*

   **Corrected**

6. *P2 L26 Remove (Weston et al 2013) from this sentence. All of the studies mentioned above show this result.*

   **Corrected**

7. *P2 L29 "(increase of the errors. . .)" The errors of what?*

   **Corrected**

8. *P3 L3 and other places. This should read "Desroziers method", not "Desroziers statistics"*

   **Corrected**

9. *P3 L7 Why not mention Section 5 in this paragraph?*

   **Corrected**
   'Then, the impact on data assimilation is reported in section 4 and validation against independent data is discussed in section 5'

10. *P4 L10 "aerosols" should be singular.*

**Corrected**

11. *P4 L18 What data file do you mean? I don't see one defined.*

   **Corrected**

Indeed, we did not mean here by previously that it was cited in the paper. We meant by 'previously' that is given as an input to the experiment.

We have modified this sentence.

12. *P4 L23 "evaluate the impact of the estimated observation error" and the error covariances, correct?*

   **Corrected**
   'evaluate the impact of the estimated observation error covariances on the analysis'

13. *P4 L26 Change "reminded" to "given"*

   **Corrected**

14. *P6 L13 Change this sentence to "...carried by a radiosonde continuously transmitting measurements as it ascends."*

   **Corrected**

15. *P6 L26 Change "found out" to "found"*

   **Corrected**

16. *P7 L26 Change "radiative transfer may" to "radiative transfer model may" and "statistics of error from the instruments" to "error statistics from instrument"*

   **Corrected**

17. *P7 L29 The second term in the expected value should be a vector transpose.*

   **Corrected**

18. *P8 L7 "(with a standard. . ." there is no closing parenthesis*

   **Corrected**

19. *P8 L12 Change "assumed" to "applied"*

   **Corrected**

20. *P8 L15 Change "An other" to "Another"*

**Corrected**

21. *P8 L 20 Change "The resulted standard deviation was greater than the one" to "The resulting standard deviations were greater than those"*

**Corrected**

22. *P9 L 22 Figure 3 shows standard deviations, not differences.*

**Corrected**
This figure was omitted in this version of paper.

23. *P14 L1-2 These two sentences are unnecessary*

**Corrected:** sentence omitted

*P15 L25 This sentence should be a part of the previous paragraph.*

**Corrected**

Auligné T, McNally AP, Dee DP. 2007. Adaptive bias correction for satellite data in a numerical weather prediction system. *Q. J. R. Meteorol. Soc.* **133**: 631–642

Bormann, N., Bonavita, M., Dragani, R., Eresmaa, R., Matricardi, M., and Mcnally, T.: Observations Through an Updated Observation Error Covariance Matrix, 2015.

Coopmann, O., Guidard, V., Fourrié, N., Josse, B., and Marécal, V.: Update of Infrared Atmospheric Sounding Interferometer (IASI) channel selection with correlated observation errors for numerical weather prediction (NWP), Atmospheric Measurement Techniques, 13, 2659– 2680, https://doi.org/10.5194/amt-13-2659-2020, 2020.

Dragani R, McNally AP. 2013. Operational assimilation of ozone-sensitive infrared radiances at ECMWF. *Q. J. R. Meteorol. Soc.* **139**: 2068–2080. DOI:10.1002/qj.2106

Emili, E., Barret, B., Le Flochmoën, E., and Cariolle, D.: Comparison between the assimilation of IASI Level 2 retrievals and Level 1 radiances for ozone reanalyses, Atmospheric Measurement Techniques Discussions, pp. 1–28, https://doi.org/10.5194/amt-2018-426, 2019.

Han W, McNally AP. 2010. The 4D-Var assimilation of ozone-sensitive infrared radiances measured by IASI. *Q. J. R. Meteorol. Soc.* **136**: 2025 – 2037.

Lefevre, F., Brasseur, G. P., Folkins, L., Smith, A. K., and Simon, P. (1994). Chemistry of the 1991-1992 stratospheric winter : Three- dimensional model simulations. *Journal of Geophysical Research*, 99(D4):8183– 8195.

Liu, D. C. and Nocedal, J.: On the limited memory BFGS method for large scale optimization, Math. Program., 45, 503–528, https://doi.org/10.1007/BF01589116, 1989

Massart, S., Piacentini, A., and Pannekoucke, O.: Importance of using ensemble estimated background error covariances for the quality of atmospheric ozone analyses, Quarterly Journal of the Royal Meteorological Society, 138, 889–905, https://doi.org/10.1002/qj.971, 2012.

Peiro, H., Emili, E., Cariolle, D., Barret, B., and Le Flochmoën, E.: Multi-year assimilation of IASI and MLS ozone retrievals: Variability of tropospheric ozone over the tropics in response to ENSO, Atmospheric Chemistry and Physics, 18, 6939–6958,

Saunders, R., Hocking, J., Turner, E., Rayer, P., Rundle, D., Brunel, P., Vidot, J., Roquet, P., Matricardi, M., Geer, A., Bormann, N., and Lupu, C.: An update on the RTTOV fast radiative transfer model (currently at version 12), Geoscientific Model Development, 11, 2717–2737, https://doi.org/10.5194/gmd-11-2717-2018, 2018.

Stewart, L. M., Dance, S. L., Nichols, N. K., Eyre, J. R., and Cameron, J.: Estimating interchannel observation-error correlations for IASI radiance data in the Met Office system, Quarterly Journal of the Royal Meteorological Society, 140, 1236–1244, https://doi.org/10.1002/qj.2211, 2014

Stockwell, W. R., Kirchner, F., Kuhn, M., and Seefeld, S. (1997). A new mechanism for regional atmospheric chemistry modeling. *Journal of Geophysical Research*, 102:25847–25879.

Tabeart, J. M., Dance, S. L., Lawless, A. S., Migliorini, S., Nichols, N. K., Smith, F., and Waller, J. A.: The impact of using reconditioned correlated observation-error covariance matrices in the Met Office 1D-Var system, Quarterly Journal of the Royal Meteorological Society, pp. 1–22, https://doi.org/10.1002/qj.3741, 2020a.

Waller, J. A., Ballard, S. P., Dance, S. L., Kelly, G., Nichols, N. K., and Simonin, D.: Diagnosing horizontal and inter-channel observa- tion error correlations for SEVIRI observations using observation-minus-background and observation-minus-analysis statistics, Remote Sensing, 8, https://doi.org/10.3390/rs8070581, 2016.

---

## Referee Report (RR1)

The authors have worked diligently to address my questions and comments, leading to a much improved manuscript. The article has greater focus and structure. I still have a few concerns, however, outlined below.

General Comments

I previously inquired about the final condition number of the R matrix and in response learned that it is around 8E6. I understand that the authors took care to ensure that the inverse of R was properly computed. However, with such a large condition number I worry that the IASI observations could be strongly down-weighted, and I wonder how the cost function in RfullExp compares to that in RdiagExp. Is it possible to include a plot of the cost function versus iterations for a cycle of RfullExp and RdiagExp, either in the response or in the revised article?

Both the other referee and I raised concerns about statistical significance in these experiments. I do not feel that an adequate response was given. To claim that results are "significant" requires a statistical analysis, for example, computing a confidence interval around the difference between the zonal averages of two experiments.

Specific Comments:

Page 2 line 24 and elsewhere: There is another recent work that can be cited here, Bathmann and Collard 2020. It might be worthwhile and relevant to include this reference as the authors also examined IASI error correlation matrices over land and sea, and assimilated IASI ozone channels.

Page 9 last paragraph and first paragraph of page 10: Where do you remark that the estimated standard deviation is proportional to the radiance values? I think the larger standard deviations in the SST channels (compared to ozone channels and in general) can probably be attributed to greater sensitivity to emissivity and cloud detection error, as well as greater representivity error.

Section 4.3 The conclusion that I can draw from this section is that larger error variances increase the convergence rate of the minimization algorithm. It is mentioned that the diagonal matrix pulls the analysis solution closer to the observations, and I think the discussion can refer back to Fig 1. The errors are generally larger in RfullExp, so these observations are being downweighted in RfullExp.

Section 4.3 The discussion about the number of iterations that are necessary for the minimization to converge is a little confusing and I wonder if the results are robust. In the first paragraph, it is stated that it converges in 90 iterations if a non-diagonal R is used. Then in the second paragraph, it is stated that it takes more than 100, 60 and 70 iterations to converge with the 1st, 2nd and 3rd estimates of R. Where did 90 come from? Also, are these numbers of iterations averaged over multiple assimilation cycles, or are they just from one cycle?

Page 16 line 8: How many ozonesonde observations are available in the high latitudes? Are there enough to quantify the significance of these results?

Technical Comments:

Page 1 line 5: "adopted in some" should be "adopted in many"

Page 1 line 16: "and in the climate" should be "and in climate"

Page 1 line 23: "component of the observation's network" should be "component of an observational network"

Page 2 line 4: after the colon, this sentence is not grammatically correct. Furthermore, parameters and climate change are not applications. Estimation of parameters and climate change studies are applications.

Page 2 line 5 and elsewhere: change MetopA to Metop-A

Page 2 line 7: can "stratosphere layer" be changed to "stratosphere"?

Page 2 line 9" change "construct more accurate" to "construct a more accurate"

Page 2 line 12 and elsewhere: change "chemistry transport model (CTM)" to just "CTM". CTM was introduce at line 8.

Page 2 line 23: change "some Numerical Weather Prediction (NWP) systems" to "many Numerical Weather Prediction (NWP) centers."

Page 3 line 5: this should say "using the Desroziers method"

Page 3 line 6: abbreviate CTM

Page 4 line 6: change "the TOVS instrument" to "TOVS instruments"

Page 4 line 10: "The radiative transfer…" multiple verb tenses are used in this sentence. It should probably only be in past tense.

Page 4 line 28: change "the Skin Surface Temperature (SST) and the ozone" to "Skin Surface Temperature (SST) and ozone"

Page 5, line 14 remove the last access statement

Page 6 line 1: change "transmitting continuously" to "continuously transmitting"

Page 6 line 11: change "section of the results" to "the results section"

Page 6, line 22: change "examine here exclusively" to "exclusively examine" and "as already reminded in the introduction and in the conclusion" to "as mentioned in the introduction"

Page 6 line 23 change "and correlation" to "and a correlation"

Page 7 lines 8-10. "The systematic error…" This sentence is redundant with the one that comes after it.

Page 8 line 5: change "have used the channel" to "used IASI channel"

Page 8 line 7: change "for the long" to "for a long"

Page 8 line 8: measurements should be singular

Page 8 line 12: change "we were used" to "we used"

Page 8, line 15 delete "also"

Page 8 lines 16-17: IASI is not an ozone instrument

Page 8 line 18: change "analyses'" to "analysis"

Page 8 line 22: change "accounted by" to "accounted for by"

Page 8 line 26: change "statistics of error" to "error statistics"

Page 9 line 1: "which may not always be the case" In practice it is almost never the case.

Page 9 line 15: change "from 3D-Var experiment that uses" to "from a 3D-Var experiment that used" and "1st" to "the 1st"

Page 9 line 22-23: "The differences…" there are mixed verb tenses in this sentence

Page 10 line 4-5 You can delete the date on this personal communication.

Page 12 line 10: change "the estimation and potential account of" to "estimating and accounting for"

Page 13 lines 11-13 "When the observation error…" This sentence should be in past tense.

Page 14 line16-17, Fig 5 "provided by… that of the RdiagExp and that of RfullExp" provided by the RdiagExp and RfullExp analyses, correct?

Page 15 line 6 Negative values indicate an improvement in the experiment over the control, right?

Page 16 lines 13-14: Change "In spite of its" to "In spite of the" and "remains always advantageous" to "remains advantageous"

Page 16 line 15: Delete "in this section"

Page 16 line 21 extra space before )

Page 16 line 24: "We have also discussed". Examined may be a more appropriate word since a lot of the discussion was removed.

Page 19 line 1: change "assimilate efficiently" to "efficiently assimilate"

Page 19 line 4 "from IASI ozone sensitive channels" and other channels as well

Page 19 line 17 Probably should remove "always" or at least move it before the verb

Page 19 line 23 "It should also note" this sentence is not grammatically correc

Page 19 line 27-28: "On the other hand…" This sentence seems a little awkward, perhaps it could be combined with the next one?

---

## Author Response (AR2)

**Answers to the referee's comments**

We would like to thank the referee for her/his review of our paper and for giving us the opportunity to improve it.

The referee's question is copied in italic and the answer is written in normal font.

We have added 'review' in the legend of the figures shown in this document to distinguish from the figures of the manuscript.

We copy the modified part of the paper here in the response. The added modifications are written in red.

We mean by 'old version' the previous revision of the manuscript and 'new version' the revised paper.

**General comments:**

1. *I previously inquired about the final condition number of the R matrix and in response learned that it is around 8E6. I understand that the authors took care to ensure that the inverse of R was properly computed. However, with such a large condition number I worry that the IASI observations could be strongly down-weighted, and I wonder how the cost function in RfullExp compares to that in RdiagExp. Is it possible to include a plot of the cost function versus iterations for a cycle of RfullExp and RdiagExp, either in the response or in the revised article?*

Indeed, the condition number is large. To compare the behavior of the cost function in RfullExp to that in RdiagExp, we present in Figure 1_review three assimilation cycles picked arbitrarily after 15 days of assimilation for both experiments: 15/07/2010 12-13h UTC, 20/07/2010 05-06h UTC, and 27/07/2010 09-10h UTC.

The six cycles (three for each experiment) are shown on separate plots because the absolute values of the cost function are not comparable among cycles and experiments. We selected three dates and different times to show that the behavior of the minimizer's iterations is somehow systematic and not scene-dependent. The plots are shown only here and not reported in the revised paper for conciseness. However, the revised article was modified to partly include the following discussion.

Figure 1_review and Figure 2_review show the cost function versus the number of iterations for the 3 cycles, for RdiagExp and for RfullExp respectively.

In the RfullExp case, the minimizer converges after almost 90 iterations (89 iterations in average over the entire month), whereas it exceeds the maximum threshold (150 iterations) in the case of RdiagExp. The two convergence criteria used in the LBFGS minimizer are based on the reduction of the cost function and of the norm of the gradient to values below typically small thresholds ( 1.e-9 for the accuracy of the reduction of the cost function between to iterations and 1.e-3 for the gradient). We remind that the limit of 150 iterations was set to save computational time. Hence, within the RdiagExp the minimization does not achieve a full convergence. However, the further reduction of the cost function during the final iterations is quite small compared to the overall reduction. As a consequence, letting the minimizer reach the full convergence (after about 200 iterations) does not affect the O3 analysis significantly (not shown). For the RfullExp, the convergence is achieved due to the stationarity of the cost function (1st criterion). The fact that the observations are downweighed in RfullExp is likely the reason for the faster convergence.

This part L13 to L15 P13 of the old version of the paper:

"In fact, the introduction of the estimated R reduces the number of iterations from 150 (a fixed value to stop iterations if the convergence criteria were not achieved to save computational time) to 90 iterations in average. This means that the CPU time is reduced by more than 150% for each assimilation cycle."

is replaced by:

"In fact, the introduction of the estimated R reduces the number of iterations from 150 (a fixed value to stop iterations if the convergence criteria were not attained to save computational time) to 89 iterations in average. This means that the CPU time is reduced by more than 150% for each assimilation cycle. The convergence criteria of the LBFGS algorithm is based on either the reduction of the cost function or the norm of its gradient below some given small thresholds. For the RfullExp, the convergence is achieved due to the stationarity of the cost function (1st criterion). The widespread correlations (high condition number) and larger variance of the estimated R matrix conduct to a downweight of the observations and are likely the reason for the improved convergence in RfullExp." in the new version of the paper L1 to L5 P14.

[Figure]

Figure 1_review: the cost function versus iterations for 15/07/2010 12-13h UTC (a), 20/07/2010 05-06h UTC (b), and 27/07/2010 09-10h UTC (c) for RdiagExp.

[Figure]

Figure 2_review: the cost function versus iterations for 15/07/2010 12-13h UTC (a), 20/07/2010 05-06h UTC (b), and 27/07/2010 09-10h UTC (c) for RfullExp.

2. *I do not feel that an adequate response was given. To claim that results are "significant" requires a statistical analysis, for example, computing a confidence interval around the difference between the zonal averages of two experiments.*

Indeed, we used the word 'significant' in the paper as well as in the previous revision. With 'significant' we meant that accounting for a more realistic observation-error estimation brought a 'remarkable' improvement in terms of results. Obtaining remarkable improvement in the validation against three independents observation networks (OMI, MLS and Ozonesondes) made us conclude that the results were 'significant'. However, the referee is right. To claim the differences between the two experiments are significant needs a statistical analysis. We present below (Figure 3_review) a t-test to evaluate the statistical significance of the differences between the two experiments in terms of the zonal averages reported in Figure 3 of the paper.

Figure 3_review reports results of Student's t-test comparing the zonal averages of the two experiments (RfullExp and RdiagExp). In fact, the zonal averages (whose differences are shown in Figure 3 of the paper) are obtained by averaging the analysis over the month of the study and over longitudes (allowing us to have a sample of size of 24(hours)x30(days)x180(longitudes)). We have used the standard deviation computed for each average to perform our test. Regions with green color reports the null hypothesis $H_0$ (the two experiments are not significantly different in terms of zonal averages) and red color report the alternative hypothesis $H_1$ (the results are significantly different in terms of zonal averages). The results shown below are obtained at 0.05 level of significance. We notice that the majority of regions report significant differences. Moreover, the regions where the differences are large in Figure 3 of the paper (between 300 hPa and 10 hPa) are statistically significant as it is shown in Figure 3_review.

Another test of the significance of the differences of the analyses with respect to the MLS and ozonesoundings measurements is reported in the question 5 of the specific comments.

We would like also to remind that our main objective was to assess the update of observation-error covariances on the assimilation results. For this, we kept the same period (one month) and system configuration already discussed in the literature (Emili et, al. 2019). Nevertheless, accounting for a long period is important to assess the robustness of the approach for a potential operational implementation. Emili et, al. 2020 have used an estimated R-matrix (as in our paper) to assess the impact of IASI measurements on global ozone reanalyses for a duration of one year (manuscript already submitted to Geoscientific Model Development). The results were similar in terms of the covariance estimation (strong correlations) and on the impact on the assimilation results (improvement of the reanalysis over the considered year). This suggests that the presented results are robust and can be extrapolated to other periods.

This discussion was added to the paper.

This part L32 P11 to L1 P12 of the old version of the paper:

'On the other hand, an important reduction of ozone is observed in the tropics at 20 hPa (more than 600 ppbv). To better...'

is replaced by:

'On the other hand, a large reduction of ozone is observed in the tropics at 20 hPa (more than 600 ppbv). We have performed a t-test to evaluate the significance of these differences between the two experiments in terms of zonal averages. These were obtained by averaging the analysis over the month of the study and over longitudes. We have used the standard deviation computed for each average to perform our test. We have noticed that in the majority of regions, especially where the differences are

large (between 300 hPa and 10 hPa), the differences are statistically significant (not shown). To better understand the impact of the estimated…'

in the new version of the paper L33 P11 to L4 P12.

[Figure]

Figure 3_review: T-test of the zonal averages of the two experiments (RfullExp and RdiagExp); the green color reports the null hypothesis $H_0$ (the two experiments are not significantly different in terms of zonal averages) and red color reports the alternative hypothesis $H_1$ (The results are significantly different in terms of zonal averages).

**Specific comments:**

1.  *Page 2 line 24 and elsewhere: There is another recent work that can be cited here, Bathmann and Collard 2020. It might be worthwhile and relevant to include this reference as the authors also examined IASI error correlation matrices over land and sea, and assimilated IASI ozone channels.*

    Indeed, it is a relevant reference to be cited here. we have added it to the manuscript

2.  *Page 9 last paragraph and first paragraph of page 10: Where do you remark that the estimated standard deviation is proportional to the radiance values? I think the larger standard deviations in the SST channels (compared to ozone channels and in general) can probably be attributed to greater sensitivity to emissivity and cloud detection error, as well as greater representivity error.*

We show in Figure 4_review the **R** standard deviation, the average of observations, and the average of the background in the observation space ($H(x_b)$). At first glance, we notice that the estimated standard deviation has a very similar shape to that of the observed radiances or the equivalent of the background in the observation space. The ratio of the estimated standard

deviation over the observation is about 5 % for SST channels and 2 % for ozone channels. We have suggested in the paper that the larger absolute error in the SST band compared to the ozone channels might be explained by the large values of the observation and the background for the SST channels in comparison with respect to the ozone channels. The (Desroziers) statistics are computed by multiplying the observation minus background values times the observation minus analysis. Since larger absolute values of **y** correspond generally to larger deviations, the observed spectral behavior of the errors seems natural. As the referee has suggested, the slightly larger relative error in the SST band could also be attributed to greater sensitivity to emissivity and representivity error. However, it is difficult to draw such conclusions from the sole Desroziers estimation procedure and we consider that a more detailed analysis of the individual sources of errors is required to better assess the effect of the different errors.

The paper was modified to include this comment.

This part of the old version of the paper L34 P9 to L2 P10:

'We remarked that the estimated standard … for the entire spectral window (not shown)'

Was replaced by:

'We have plotted the R standard deviation, the average of observations, and the average of the background in the observation space on the same figure (not shown).  We have noticed that the estimated standard deviation has a very similar shape to that of the observed radiances or the equivalent of the background in the observation space. This may suggest that the larger absolute error in the SST band compared to the ozone channels might be explained by the large values of the observation and the background for the SST channels in comparison with respect to the ozone channels. It could also be attributed to greater sensitivity to emissivity and representivity error.'

This part was added to the new version of the paper L8 to L13 P10:

[Figure]

Figure 4_review: Estimated standard deviation (black) on the left axis, and observed radiances (red) and the equivalent of the background averaged over the month (blue) on the right axis.

3.    *Section 4.3 The conclusion that I can draw from this section is that larger error variances increase the convergence rate of the minimization algorithm. It is mentioned that the diagonal matrix pulls the analysis solution closer to the observations, and I think the discussion can refer back to Fig 1. The errors are generally larger in RfullExp, so these observations are being downweighed in RfullExp.*

Indeed, in the case of RfullExp the errors are larger and observations are downweighted as a result. We added a reference to the Figure 1 in the discussion.

This part of the paper was modified (see comment 1 of 'general comments')

*Section 4.3 The discussion about the number of iterations that are necessary for the minimization to converge is a little confusing and I wonder if the results are robust. In the first paragraph, it is stated that it converges in 90 iterations if a non-diagonal R is used. Then in the second paragraph, it is stated that it takes more than 100, 60 and 70 iterations to converge with the 1st, 2nd and 3rd estimates of R. Where did 90 come from? Also, are these numbers of iterations averaged over multiple assimilation cycles, or are they just from one cycle?*

Indeed, the way we have presented the number of iterations was not done properly and may create confusion. In fact, we have picked arbitrarily the number of iterations (90) from an assimilation cycle we presented in the first paragraph. In the second paragraph, we have changed the assimilation cycle considered. To correct this, we have averaged the total of iterations over all cycles.

We found that the first estimation needs an average of 149 iterations to converge whereas the second estimation requires only an average of 89 iterations.

The paper was modified to include this discussion.

This part of the new version of the paper was corrected L14 P14:

'The minimizer takes 149 iterations in average to converge. (average computed for all the assimilation cycles of the month). We used the analysis given by the 1$^{st}$ experiment to estimate another R-matrix. We have used this estimation to run another assimilation cycle (2$^{nd}$ experiment). We have noticed that the minimizer needs about 89 iterations in average to converge. We have modified the R-matrix of the 1$^{st}$ experiment by keeping its correlations and replacing its standard deviation with that of R used in the 2$^{nd}$ experiment. The resulting matrix was used to run a 3$^{rd}$ assimilation experiment. The minimizer needs less than 90 iterations to converge. The results of the 3$^{rd}$ experiment seem to suggest that updating the variance has a larger impact on the convergence speed.'

4. *Page 16 line 8: How many ozonesondes observations are available in the high latitudes? Are there enough to quantify the significance of these results?*

*4.1.      How many ozonesondes observations are available in the high latitudes?*

We have used 219 radiosoundings whose geographical distribution is presented in Figure 5_review.

[Figure]

Figure 5_review: Geographical distribution of used ozone soundings.

*4.2.      Are they enough to quantify the significance of the results?*

As we have mentioned in the comment 2 of general comments, we meant by 'significant' that accounting for a more realistic observation-error estimation brought a 'remarkable' improvement in terms of results. However, quantifying the statistical significance requires statistical analysis.

To discuss this question, we have applied the t-test to the differences between analyses of the two experiments and observations (ozonesondes then MLS). For each observation type (MLS and ozonesondes) and for both experiments, we have computed $H(x_a)$-observation. We have averaged the differences over the number of available observations (ozonesondes and MLS separately). The significance of these differences between the two experiments is performed with 0.05 level confidence.

In other words, we have performed the t-test for the averages (over the number of observations) of these two quantities:

$$\varepsilon^{RdiagExp} = H(x_{a\_RdiagExp}) - Y. \qquad and \qquad \varepsilon^{RfullExp} = H(x_{a\_RfullExp}) - Y$$

with $x_{a\_RdiagExp}$ is the analysis from the RdiagExp, $x_{a\_RfullExp}$ the analysis from the RfullExp, H the observation operator and Y stands for observations (ozonesondes or MLS).

We present below the results and the number of used observations for ozonesoundings (Figure 6_review) and for MLS (Figure 7_reveiw). H0 stands for the null hypothesis, the averages of the analysis minus observations are not significantly different between RdiagExp and RfullExp. It is set to 'False' (red points) when the differences are statistically significant and 'True' (green points) for the inverse. The levels where the observations are not available are shown in blue points.

We notice, in Figure 6_review, that the significance of the differences between the two experiments' analyses and the ozonesoundings differs over the levels. The question raised by the referee is relevant. In fact, the conclusion we have made 'significant results' has to be discussed in detail (as function of levels). The reduction of the error between 20 and 50 hPa, and between 300 and 400 hPa reported in Figure 6 of the paper (all) is significant. For the low troposphere the differences are not statistically significant.

To complete this discussion, we present the MLS results in Figure 7_review. Unlike the ozonesoundings results, the differences with respect to the MLS are statistically significant for the all levels discussed in the paper (between 10 and 170 hPa, showed in Figure 6 of the paper (all)).

In conclusion, we admit that ozonesoundings as a single source of information is not sufficient to quantify the significance of the differences noted when accounting for an updated observation error. However, accounting for other sources of information (OMI and MLS) that report the improvement of the results encountered in the radiosoundings validation combined with the significance of MLS over the considered levels suggests that our results are significant.

We have modified the revised paper to include this discussion.

This part added to the new version L12 P16:

'To evaluate the significance of the differences between the analyses of the two experiments with respect to MLS and ozonesoundings measurements, we have performed the t-test of the differences between analyses and observations (ozonesondes then MLS). We have noticed that for the ozonesoundings, the significance differs among vertical levels. The reduction of the error between 20 and 50 hPa, and between 300 and 400 hPa reported in Figure 6 is statistically significant. For the low troposphere the differences are not significant. Unlike the ozonesoundings results, the differences with respect to the MLS measurements are statistically significant for all levels discussed in MLS validation.'

[Figure]

[Figure]

Figure 6_review: (a) T-test of the averages of the analyses minus observations (given by the ozonesoundings) of the two experiments (RfullExp and RdiagExp); the green points report the null hypothesis $H_0$ (the averages are not significantly different) and red points report the alternative hypothesis $H_1$ (the averages are significantly different), and the blue show the levels where observations are not available. The number of used observations is shown on (b).

[Figure]

[Figure]

Figure 7_review: (a) T-test of the averages of the analyses minus observations (given by MLS) of the two experiments (RfullExp and RdiagExp); the green points report the null hypothesis $H_0$ (the averages are not significantly different) and red points report the alternative hypothesis $H_1$ (the averages are significantly different), and the blue show the levels where observations are not available. The number of used observations is shown on (b).

**3. Technical comments:**

*Page 1 line 5: "adopted in some" should be "adopted in many":*

**Corrected**

*Page 1 line 16: "and in the climate" should be "and in climate"*

**Corrected**

*Page 1 line 23: "component of the observation's network" should be "component of an observational network"*

**Corrected**

*Page 2 line 4: after the colon, this sentence is not grammatically correct. Furthermore, parameters and climate change are not applications. Estimation of parameters and climate change studies are applications.*

**Corrected**

*Page 2 line 5 and elsewhere: change MetopA to Metop-A*

**Corrected**

*Page 2 line 7: can "stratosphere layer" be changed to "stratosphere"?*

**Corrected**

*Page 2 line 9" change "construct more accurate" to "construct a more accurate"*

**Corrected**

*Page 2 line 12 and elsewhere: change "chemistry transport model (CTM)" to just "CTM". CTM was introduce at line 8.*

**Corrected**

*Page 2 line 23: change "some Numerical Weather Prediction (NWP) systems" to "many Numerical Weather Prediction (NWP) centers."*

**Corrected**

*Page 3 line 5: this should say "using the Desroziers method"*

**Corrected**

*Page 3 line 6: abbreviate CTM*

**Corrected**

*Page 4 line 6: change "the TOVS instrument" to "TOVS instruments"*

**Corrected**

*Page 4 line 10: "The radiative transfer..." multiple verb tenses are used in this sentence. It should probably only be in past tense.*

**Corrected**

*Page 4 line 28: change "the Skin Surface Temperature (SST) and the ozone" to "Skin Surface Temperature (SST) and ozone"*

**Corrected**

*Page 5, line 14 remove the last access statement*

**Corrected**

*Page 6 line 1: change "transmitting continuously" to "continuously transmitting"*

**Corrected**

*Page 6 line 11: change "section of the results" to "the results section"*

**Corrected**

*Page 6, line 22: change "examine here exclusively" to "exclusively examine" and "as already reminded in the introduction and in the conclusion" to "as mentioned in the introduction"*

**Corrected**

*Page 6, line 23 change "and correlation" to "and a correlation"*

**Corrected**

*Page 7 lines 8-10. "The systematic error…" This sentence is redundant with the one that comes after it.*

**Corrected**

*Page 8 line 5: change "have used the channel" to "used IASI channel"*

**Corrected**

*Page 8 line 7: change "for the long" to "for a long"*

**Corrected**

*Page 8 line 8: measurements should be singular*

**Corrected**

*Page 8 line 12: change "we were used" to "we used"*

**Corrected**

*Page 8, line 15 delete "also"*

**Corrected**

*Page 8 lines 16-17: IASI is not an ozone instrument*

**Corrected**

*Page 8 line 18: change "analyses" to "analysis"*

**Corrected**

*Page 8 line 22: change "accounted by" to "accounted for by"*

**Corrected**

*Page 8 line 26: change "statistics of error" to "error statistics"*

**Corrected**

*Page 9 line 1: "which may not always be the case" In practice it is almost never the case. Page 9 line 15: change "from 3D-Var experiment that uses" to "from a 3D-Var experiment that used" and "1st" to "the 1st"*

**Corrected**

*Page 9 line 22-23: "The differences..." there are mixed verb tenses in this sentence*

**Corrected**

*Page 10 line 4-5 You can delete the date on this personal communication.*

**Corrected**

---

## Author Response (AR3)

Answer to the Editor

First of all, we would like to thank the Editor for accepting our paper for publication.

We copy here the question of the Editor in italic font. The answer follows in normal font.

Comment 1:

*Although you answered the referee regarding the question of the cost function (see below), it was not clear to me if the RfullExp and RdiagExp options actually converge on the same retrieval solution (to within the expected formal retrieval errors, although you do indicate that the RfullExp solution "generally larger errors"). However, I assume you tested this and therefore please clarify in the final m/s e.g. in Fig 6 "all" around 200 hPa are the blue and green lines separated by more than the RMS of their combined retrieval errors?*

Answer 1:

In our 3D data assimilation framework, the 'retrieval solution' corresponds to a global ozone 3D analysis. RfullExp and RdiagExp do not converge to the same 3D analysis since the two analyses are the result of two different minimization problems (i.e. based on different input error covariances).

It is worth reminding that we have discussed two aspects in our paper: the estimation of the R-matrix (interchannel error covariances) and its impact on the assimilation results. Hence, two type of errors were discussed: the observation errors (i.e. the R-matrix, further used as input to compute *RfullExp*) and the error of the *RfullExp* analysis.

Concerning the observations error, the estimated R-matrix shows larger errors in terms of standard deviation (Fig 1) and interchannel error correlations (Fig 2) comparing with the diagonal R-matrix (lower standard deviation and neglected interchannel error-correlations) used in RdiagExp. We referred to this when we employed the expression 'generally larger errors' (In our answer to the comment 3 of the referee, specific comments in the latest revision).

Concerning the analysis error, it is not feasible to compute and store the analysis' error covariance in the framework of 3D-Var due to its huge size. This is somewhat different with respect to satellites 'retrievals', where 1D algorithms are used and retrievals errors are generally delivered as output. Therefore, to quantify the impact of using an estimated matrix, we have compared our results to independent data and with respect to a free run (no assimilation) for both experiments (RfullExp and RdiagExp). The use of an estimated R-matrix (the case of RfullExp) has remarkably reduced the error of the analysis in the stratosphere comparing to RdiagExp (Fig 6). Around 200 hPa, the RdiagExp analysis error is smaller than RfullExp (Fig 6 'all'). However, we cannot verify whether this difference sits within the correspondent analysis ('retrieval') errors, since we have no access to their values.

Comment 2:

*In the above revised text, I suggest replacing 'conduct' with 'lead'.*

Answer 2:

Corrected.